# Redundant and non-redundant cytokine-activated enhancers control *Csn1s2b* expression in the lactating mouse mammary gland

Hye Kyung Lee 🄳 1✉, Michaela Willi¹, Tyler Kuhns¹, Chengyu Liu² & Lothar Hennighausen 🄳 1✉

Enhancers are transcription factor platforms that synergize with promoters to control gene expression. Here, we investigate enhancers that activate gene expression several hundred-fold exclusively in the lactating mouse mammary gland. Using ChIP-seq for activating histone marks and transcription factors, we identify two candidate enhancers and one super-enhancer in the *Csn1s2b* locus. Through experimental mouse genetics, we dissect the lactation-specific distal enhancer bound by the mammary-enriched transcription factors STAT5 and NFIB and the glucocorticoid receptor. While deletions of canonical binding motifs for NFIB and STAT5, individually or combined, have a limited biological impact, a non-canonical STAT5 site is essential for enhancer activity during lactation. In contrast, the intronic enhancer contributes to gene expression only in late pregnancy and early lactation, possibly by interacting with the distal enhancer. A downstream super-enhancer, which physically interacts with the distal enhancer, is required for the functional establishment of the *Csn1s2b* promoter and gene activation. Lastly, NFIB binding in the promoter region fine-tunes *Csn1s2b* expression. Our study provides comprehensive insight into the anatomy and biology of regulatory elements that employ the JAK/STAT signaling pathway and preferentially activate gene expression during lactation.

¹ Laboratory of Genetics and Physiology, National Institute of Diabetes and Digestive and Kidney Diseases, US National Institutes of Health, Bethesda, MD 20892, USA. ² Transgenic Core, National Heart, Lung, and Blood Institute, US National Institutes of Health, Bethesda, MD 20892, USA. ✉email: hyekyung.lee@nih.gov; lotharh@niddk.nih.gov

Enhancers are transcription component platforms that control the location, timing and intensity of gene expression[1,2]. While current approaches, such as the ChIP-seq and physical contact studies, are useful in identifying candidate enhancers, their biological predictions are limited and validation through genetic experiments is needed. Enhancers are occupied by multiple transcription factors (TFs) that might bind directly to DNA through their respective recognition motifs or indirectly through tethering. Since experimental genetic studies generally ablate the entire enhancer, the structural and functional contribution of individual TFs remains to be understood.

Several hundred genes are uniquely expressed in mammary tissue and activated by pregnancy and lactation hormones through the tyrosine kinase JAK2 and the transcription factors Signal Transducer and Activator of Transcription (STAT) 5A and 5B (referred to as STAT5)[3–5]. STAT5 is activated by prolactin and it controls mammary alveolar development during pregnancy and the activation of genetic programs resulting in lactation[3,4]. While most STAT5 target genes are highly induced during pregnancy and to a lesser extent during lactation[6], the activation of the *Csn1s2b* gene[7] occurs preferentially during lactation[8], possibly through enhancers that are specifically established after parturition[8]. ChIP-seq profiles for STAT5 and H3K27ac and other mammary-enriched TFs suggested the presence of highly complex mammary enhancers[8]. Although most of these enhancers appear to depend on STAT5 as the anchor for the establishment of larger protein complexes, the stage-specific generation of enhancers remains to be understood. It is not known why seemingly structurally identical enhancers can be activated by pregnancy hormones either during pregnancy or lactation.

Caseins, the major components of milk, are cardinal proteins that are unique to mammals. They are evolved from secretory calcium-binding phosphoproteins (SCPP) with the odontogenic ameloblast–associated (*ODAM*) gene being possibly a founding member. While a CSN3-like protein is already found in early amniotes and appears to be the first member of the family of five caseins, CSN1s2b is a more recent addition that evolved through gene duplication[9]. The mouse casein locus spans ~400 kbp and consists of five casein genes (*Csn1s1*, *Csn2*, *Csn1s2a*, *Csn1s2b* and *Csn3*) and at least three SCPP genes (*Prr27*, *Odam* and *Fdcsp*) that are expressed in salivary glands and possibly other secretory tissues. The casein locus remains a fertile ground for exploring tissue-restricted and hormone-controlled gene regulation. Foremost, while the casein genes are expressed exclusively in mammary tissue and are induced by pregnancy and lactation hormones, the interspersed genes are expressed preferentially in salivary gland tissue. Among the five casein genes, expression of *Csn1s2b* is uniquely different from the other four in that its activation predominantly occurs during lactation and not during pregnancy. The evolution of the five caseins through gene duplication begs the question to what extent regulatory elements were duplicated, developed de novo or even shared between genes. It seems plausible that regulatory elements controlling the ancient SCCP genes were repurposed and acquired features that permitted their activity in secreting mammary gland cells. It also remains to be determined whether regulatory elements controlling the ancestral *Csn3* gene were acquired by the younger *Csn1s2b* gene, which is separated from *Csn3* by three SCCP genes.

Here, we used ChIP-seq for activating histone marks and transcription factors to identify candidate enhancers in mammary tissue during pregnancy and lactation. We identified two candidate enhancers and one super-enhancer in the extended *Csn1s2b* locus and investigated a potential synergy between the prolactin-induced TF STAT5 and the mammary-enriched Nuclear Factor I B (NFIB) in the establishment of lactation-specific regulatory elements. For this, we employed experimental mouse genetics and functionally dissected the two enhancers, the super-enhancer and the *Csn1s2b* promoter. This permitted us to define the contributions of individual enhancers and the significance of STAT5 and NFIB in the activating the *Csn1s2b* gene during pregnancy and lactation.

## Results

**A *Csn1s2b* distal enhancer is activated in mammary tissue during lactation.** The five casein genes, positioned within a ~400 kbp locus, are expressed exclusively in mammary tissue under the control of pregnancy and lactation hormones (Supplementary Table 1). Interspersed in this locus are three genes that are preferentially expressed in salivary glands. While four out of the five casein genes are highly induced during pregnancy, *Csn1s2b* is activated preferentially, and up to several-hundred-fold, during lactation, suggesting the presence of distinct regulatory elements. The adjacent *Csn1s2a* and *Csn1s2b* genes, which arose by gene-duplication prior to the split of eutherian mammals[10,11], are subject to a different regulation. While *Csn1s2b* expression increased more than 250-fold between day 1 of lactation (L1) and day 10 (L10), *Csn1s2a* expression increased approximately 6-fold (Fig. 1a) suggesting the presence of regulatory elements that uniquely respond to lactation stimuli, with prolactin the most prominent hormone. To identify such putative regulatory elements, we dug deeper and used ChIP-seq profiling for transcription factor binding and the presence of activating histone marks (Fig. 1b–d and Supplementary Fig. 1a–c). Binding of STAT5A/B (referred to as STAT5), transcription factors activated by prolactin, was detected at three sites upstream of the *Csn1s2a* gene and at two sites at the *Csn1s2b* gene (Fig. 1b). Each of the five sites bound by STAT5 coincided with at least one GAS motif (the sequence recognized by STAT family members) supporting a direct protein-DNA interaction. The most proximal STAT5 binding sites at the two genes are close to the TSS, suggesting that they could be part of a combined promoter-enhancer unit. While maximum STAT5 binding at the *Csn1s2a* sites was already observed at day 18 of pregnancy (p18) and remained high throughout lactation, STAT5 binding at the candidate *Csn1s2b* enhancer was marginally detectable at p18 and was fully established between L1 and L10 (Fig. 1b and Supplementary Fig. 1a–c). Pol II loading and H3K4me3 coverage at the two loci also reflects the differential expression of the two genes (Fig. 1b–d).

A candidate distal enhancer (DE) bound by STAT5 and other TFs, including the glucocorticoid receptor (GR), NFIB and MED1, was identified 2.3 kb 5′ of the *Csn1s2b* TSS (Fig. 1d). STAT5 and NFIB binding coincided with their respective recognition motifs (TTCnnnGAA for STAT5 and TGGCA/TGCCA for NFIB), suggesting direct protein-DNA interactions. One GR half site motif (TGTYCY/RGRACA)[12–14] was identified within the DE and overlapped with the GAS motif in STAT5 binding site S1 (Supplementary Table 3). Putative GR motifs were located within the promoter binding site and also in two out of the three sites at the neighboring *Csn1s2a* gene (Fig. 1c). While unbiased motif searches for the mammary-enriched TFs STAT5 and NFIB have been conducted in mammary tissue from lactating mouse, no such information was available for the GR. We therefore performed a de novo motif search using GR ChIP-seq data from L10 mammary tissue (Supplementary Fig. 2). Out of the approximately 26,000 sites bound by the GR, 22,675 coincided with H3K27ac marks, indicative of candidate regulatory elements. Motifs for transcription factors (ETS factors, STAT5 and Nuclear Factor I family) known to control mammary development and function were significantly enriched at the 22,617 sites bound by GR and marked by H3K27ac (±500 bp).

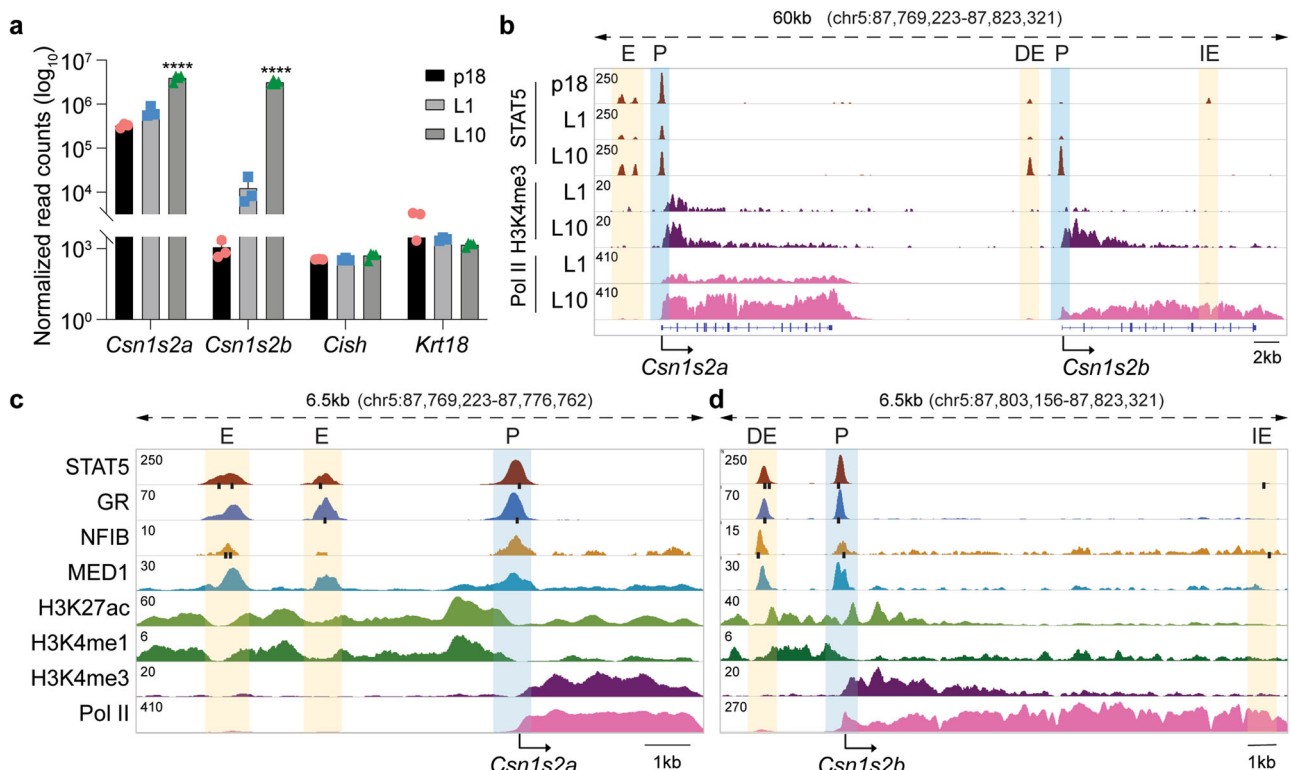

**Fig. 1 Chromatin structures and activity of the *Csn1s2a/b* locus. a** mRNA levels were measured by RNA-seq at day 18 of pregnancy (p18), day one of lactation (L1) and L10. Results are shown as the means ± s.e.m. of independent biological replicates (p18 and L1, $n = 3$; L10, $n = 4$). Two-way ANOVA followed by Tukey's multiple comparisons test was used to evaluate the statistical significance of differences. The *Cish* and *Krt18* genes served as controls. **b** ChIP-seq data for STAT5a binding, H3K4me3 marks and Pol II loading at the *Csn1s2a-Csn1s2b* locus during pregnancy (p18) and lactation (L1 and L10). The orange shades indicate candidate enhancers and the blue shades are promoter regions. The *Cish* locus served as control (Supplementary Fig. 1). DE, *Csn1s2b* distal enhancer; IE, intronic enhancer; P, promoter. **c–d** TF binding and activating histone marks at the *Csn1s2a* (**c**) and *Csn1s2b* (**d**) genes at L10. The black bars indicate DNA binding motifs for STAT5 (TTCnnnGAA), GR (RGXACAnnnTGTXCY) and NFIB (TGGCA/TGCCA). The orange and blue shades indicate putative enhancers and promoters, respectively.

GR ChIP-seq peaks were enriched for the GR half-site motif (TGTYCY/RGRACA)[14] (Supplementary Fig. 2).

STAT5 and NFIB binding at the *Csn1s2b* DE do not overlap (Figs. 1d and 2a), suggesting the possibility of their distinct contributions in establishing a functional enhancer. Conversely, STAT5 and NFIB binding coincides at the *Csn1s2a* candidate regulatory regions (Fig. 1c). The presence of H3K4me1 marks in the candidate *Csn1s2b* DE supports its status as enhancer. STAT5 binding at a GAS motif within intron 9 of the *Csn1s2b* gene was detected during pregnancy but it sharply declined during lactation (Fig. 1b and Supplementary Fig. 1b) suggesting that it might activate the locus during pregnancy.

**Identification of TF building blocks required for the establishment and function of the distal enhancer.** Next, we explored the biological significance of the *Csn1s2b* DE and its individual building blocks through the introduction of mutations into the mouse genome (Supplementary Tables 2 and 3). We addressed the potential function of the two canonical GAS motifs (TTCnnnGAA) recognized by STAT5 (sites S1 and S2) and the NFIB motif (TGGCA) (N), all of which align with the respective ChIP-seq peaks (Fig. 2a). A non-canonical GAS motif (S3) with a 4 bp spacer (TTCnnnnGAA) was detected between the NFIB site and the STAT5 site S1. Such non-canonical GAS motifs are known to be recognized by STAT6. We generated mice carrying individual or combinatorial mutations disrupting the GAS and NFIB motifs (Fig. 2a). Although the deletion of a single T from the S2 GAS motif (ΔS2) (Supplementary Table 3) led to an

insignificant reduction of *Csn1s2b* expression (Fig. 2b), it nevertheless resulted in an ~40% loss of STAT5 binding (Fig. 2c, d), suggesting a compensatory role of STAT5 binding to site S1. The *Wap* and *Cish* genes were used as ChIP-seq controls (Supplementary Fig. 3a). Disruption of the NFIB motif (ΔN) was accomplished with a 15 bp deletion that removed the 'A' from the canonical TGGCA motif (Supplementary Table 3). *Csn1s2b* expression was overtly unaffected, as was NFIB and STAT5 binding at the DE (Fig. 2b, c). Although the mutated site (TGGCT) does not match known NFIB binding sites, we cannot rule out the possibility that this site, in conjunction with intact STAT5 sites facilitates NFIB binding. To further address this issue, we introduced a 14 bp deletion spanning the entire NFIB site into the ΔS2 background (ΔN/S2). *Csn1s2b* expression was reduced by ~45% (Fig. 2b) and coincided with greatly reduced STAT5 binding and diminished H3K27ac marks (Fig. 2c, d). The STAT5 and NFIB coverage in the *Csn12b* enhancer of mutants (ΔS2, ΔN and ΔN/S2) was confirmed by the raw read mapping (Supplementary Fig. 3b), demonstrating that STAT5 and NFIB are still bound to the mutant enhancer. GR binding was reduced in the ΔS2 mutant (Supplementary Figure 4) suggesting either cooperativity between these sites or tethering of GR to STAT5.

**Importance of the non-canonical GAS motif in *Csn1s2b* gene expression.** To address the possibility of additional TF binding sites in the DE, we dug deeper and analyzed the remaining sequences under the ChIP-seq peaks. First, we introduced a deletion spanning the NFIB site and the non-canonical GAS

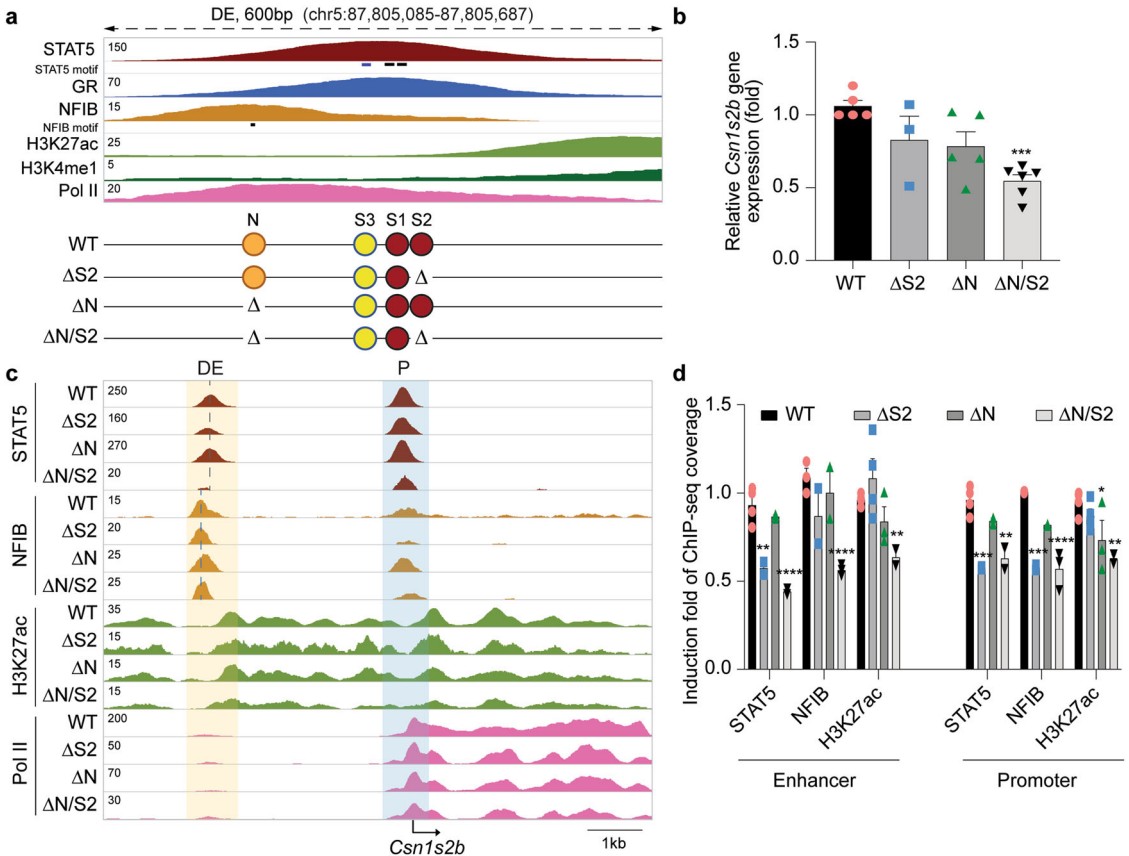

**Fig. 2 Redundant and non-redundant functions of STAT5 and NFIB sites in the *Csn1s2b* distal enhancer. a** Genomic feature of the *Csn1s2b* distal enhancer (DE) and diagram of the deletions introduced in the mouse genome using CRISPR/Cas9 genome editing. TF binding sites were mutated individually (ΔS2 and ΔN) or in combination (ΔN/S2). While S1 and S2 display canonical GAS motifs (burgundy circles), S3 (yellow circle) is a non-canonical sequence with a 4bb spacer. N (orange circle) is a NFIB binding site. **b** *Csn1s2b* mRNA levels in day 10 lactating (L10) mammary tissues from WT and mutant mice were measured by qRT–PCR and normalized to *Gapdh* levels. Results are shown as the means ± s.e.m. of independent biological replicates (WT and ΔN, $n = 5$; ΔS2 and ΔN/S2, $n = 6$). One-way ANOVA followed by Dunnett's multiple comparisons test was used to evaluate the statistical significance of differences between WT and each mutant mouse line. *p*-value = 0.21, 0.63 and 0.0006, respectively. **c** The *Csn1s2b* locus was profiled using ChIP-seq data of WT and mutant tissue. **d** STAT5, NFIB and H3K27ac coverage was calculated after variation between data set was normalized with *Cish* promoter coverage. Results are shown as the means ± s.e.m. of independent biological replicates (STAT5, H3K27ac of WT and H3K27 of ΔS2, $n = 4$; NFIB of WT, H3K27ac of ΔN and NFIB of ΔN/S2, $n = 3$; STAT5, NFIB of ΔS2 and ΔN, STAT5 and H3K27ac of ΔN/S2, $n = 2$).

motif S3 (ΔN/S3) (Fig. 3a and Supplementary Table 3). *Csn1s2b* mRNA levels declined by 86%, which paralleled a more than 70% reduction of STAT5 occupancy and H3K27ac marks at the DE and promoter (Fig. 3b–d). *Cish* and *Wap* were used as ChIP-seq controls (Supplementary Fig. 5). To determine whether sites S1 and S2 foster the residual enhancer activity, we introduced a deletion spanning S1 and S3 (Fig. 3a and Supplementary Table 3). In addition, as a result of imperfect CRISPR/Cas9 genome editing[15], the NFIB site was disrupted. *Csn1s2b* mRNA levels in this mutant (ΔN/S1/3) were reduced by ~89% (Fig. 3b) and the remaining GAS motif S2 is sufficient for residual STAT5 binding (Fig. 3c, d). Lastly, we generated mice carrying a deletion spanning site S3 and the point mutation in S2 (ΔS2/3) (Fig. 3a). *Csn1s2b* expression levels were reduced by more than 95% (Fig. 3b) coinciding with a complete absence of STAT5 and NFIB binding, despite an intact NFIB DNA binding motif (Fig. 3c, d). Similarly, no GR binding was detected despite the presence of an intact GR half-site (Supplementary Fig. 4). The reduction of H3K27ac marks coincided with reduced gene expression (Fig. 3c, d and Supplementary Fig. 5). The combined absence of sites S2 and S3 resulted in a complete absence of TF binding at the distal enhancer and also in a sharp reduction at the promoter proximal site (Fig. 3c, d), in agreement with the almost

complete loss of *Csn1s2b* expression. The STAT5 and NFIB coverage and H3K27ac marks at the *Csn1s2b* locus in wt and mutant tissues are shown in Supplementary Fig. 6. These results provide evidence that the non-canonical STAT5 binding motif, and possibly the surrounding sequences, is a key element in the DE and synergizes with the canonical site S2. The integration of the results from all of the mutants strongly suggests that STAT5 preferentially binds at the non-canonical site S3. Ultimate proof for this conclusion would require the specific deletion of S3, which we did not accomplish in this study.

**Temporal activity of the *Csn1s2b* intronic enhancer.** Our ChIP-seq data had revealed a putative enhancer in intron 9 of the *Csn1s2b* gene (Fig. 1b and Supplementary Fig. 1a-b). Like other mammary candidate enhancers, it was bound by STAT5, NFIB, GR, MED1 and Pol II and coincided with activating histone marks H3K27ac and H3K4me1 (Fig. 1d). STAT5 binding was prominent during pregnancy and declined during lactation (Fig. 4a), suggesting the possibility of a priming function in the activation of the *Csn1s2b* locus. To test this hypothesis, we generated mice with two distinct deletions targeting the GAS and NFIB motifs. The GAS motif was disrupted through the introduction of either a 3 bp or 14 bp deletion (ΔIE-S) and a 36 bp

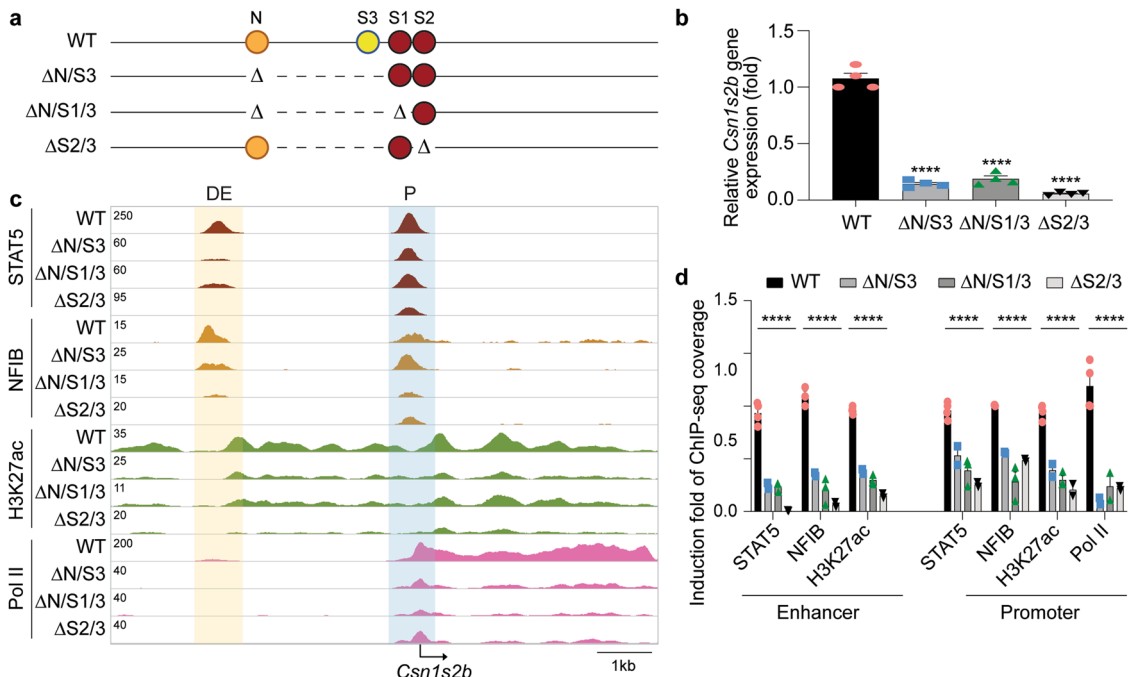

**Fig. 3 Requirement of the non-canonical STAT5 site in the *Csn1s2b* distal enhancer. a** Diagram of the enhancer deletions introduced using CRISPR/Cas9 genome editing. The non-canonical STAT5 motif was deleted in combination with other enhancer motifs (ΔN/S3, ΔN/S1/3 and ΔS2/3). The canonical GAS motifs S1 and S2 are shown as burgundy circles, and the non-canonical GAS motif S3 is shown in yellow. NFIB binding site is shown in orange. **b** *Csn1s2b* mRNA levels were measured by qRT-PCR in day 10 lactating (L10) mammary tissue of WT and mutant mice and normalized to *Gapdh* levels. Results are shown as the means ± s.e.m. of independent biological replicates (WT, n = 5; ΔN/S3, ΔN/S1/3 and ΔS2/3, n = 4). One-way ANOVA followed by Dunnett's multiple comparisons test was used to evaluate the statistical significance of differences between WT and each mutant mouse. *p*-value = 0.0001, 0.0001 and 0.0001, respectively. **c** The consequences of enhancer deletions were confirmed by STAT5A, NFIB, H3K27ac and Pol II ChIP-seq analysis in WT and mutant tissues at L10. The *Cish* locus served as control (Supplementary Fig. 3). **d** STAT5, NFIB, H3K27ac and Pol II coverage was calculated after variation between data set was normalized by *Cish* promoter coverage. Results are shown as the means ± s.e.m. of independent biological replicates (n = 2 to 4).

deletion covered the GAS and NFIB motifs (ΔIE) (Fig. 4a and Supplementary Table 3). As expected, the combined deletion of the STAT5 and NFIB binding sites (ΔIE) abrogated STAT5 binding, H3K27ac and Pol II coverage at L1 (Fig. 4b). Residual NFIB binding suggests that this TFs might bind indirectly to chromatin and not through its core DNA motif. STAT5 binding was also diminished at the DE and promoter region, indicating a functional role of the intronic enhancer. The *Cish* gene was used as a ChIP-seq control (Fig. 4b). Deletion of the GAS site by itself (ΔIE-S) or in combination with the NFIB motif (ΔIE) resulted in a reduction of *Csn1s2b* mRNA levels at L1 by ~50% and 80%, respectively (Fig. 4c). In contrast, at L10, *Csn1s2b* mRNA levels were not significantly reduced, in agreement with the absence of enhancer structures in wt mammary gland tissue. In accordance with the expression data, the ΔIE mutation impacted STAT5 binding and H3K27ac at the promoter at L1, but not so at L10 (Fig. 4d).

**A 3′ super-enhancer activates *Csn1s2b* expression.** Our genetic analyses demonstrated that the distal enhancer is a key driver of *Csn1s2b* expression throughout lactation and the intronic enhancer is most prominent at the intersection of pregnancy and early lactation. While there are no additional overt enhancer marks between the body of the *Csn1s2b* gene and the neighboring *Csn1s2a* and *Prr27* genes, we explored the possibility of further enhancers and monitored activating chromatin marks in the extended *casein* locus at L1 (Fig. 5a). A 10 kb sequence highly enriched with H3K27ac and H3K4me1 marks was identified 3′ of the *Prr27* gene, 65 kb 3′ of the *Csn1s2b* gene (Fig. 5a). ChIP-seq

identified STAT5, NFIB, GR and MED1 binding to several sites in this region and the Rose algorithm called it a super-enhancer (SE). Expression of the *Prr27* gene, which is located between the *Csn1s2b* gene and the SE, is barely detectable in mammary tissue, suggesting that it is not a genuine target of this candidate SE. 3C analyses demonstrated that this SE interacted with the neighboring *Csn1s2b* and *Csn3* genes (Fig. 5b). Deletion of the SE from the mouse genome resulted in a more than 90% reduction of *Csn1s2b* mRNA at p18 (Fig. 5c) suggesting its pivotal role in gene activation during pregnancy. Expression of the *Csn1s2a* gene was reduced by ~40%, which was statistically not significant. Since mice lacking the SE failed to nurse their pups, it was not possible to investigate *Csn1s2b* expression during lactation. Failure to lactate is not the result of reduced expression of *Csn1s2b* since mice lacking the distal enhancer can lactate despite even lower *Csn1s2b* expression levels. To identify the cause of lactation failure, further research is needed.

Deletion of the SE resulted in the complete loss of H3K27c marks and transcription factor binding at that region (Fig. 5d). Importantly, the absence of significant H3K27ac marks in the *Csn1s2b* promoter and the distal (DE) and intronic enhancers (IE) is reflective of loss of gene expression. Notably, deletion of the SE severely impacted STAT5 binding at the *Csn1s2b* promoter but less so at the DE (Fig. 5d). Conversely, STAT5 binding at the intronic enhancer was elevated. The *Csn1s2a* locus served as a ChIP-seq control. These findings strongly suggest a dominant function of the SE in activating the *Csn1s2b* promoter and that the distal and intronic enhancers cannot function independently and compensate for the absence of the SE at the transition of pregnancy to lactation (p18 and L1).

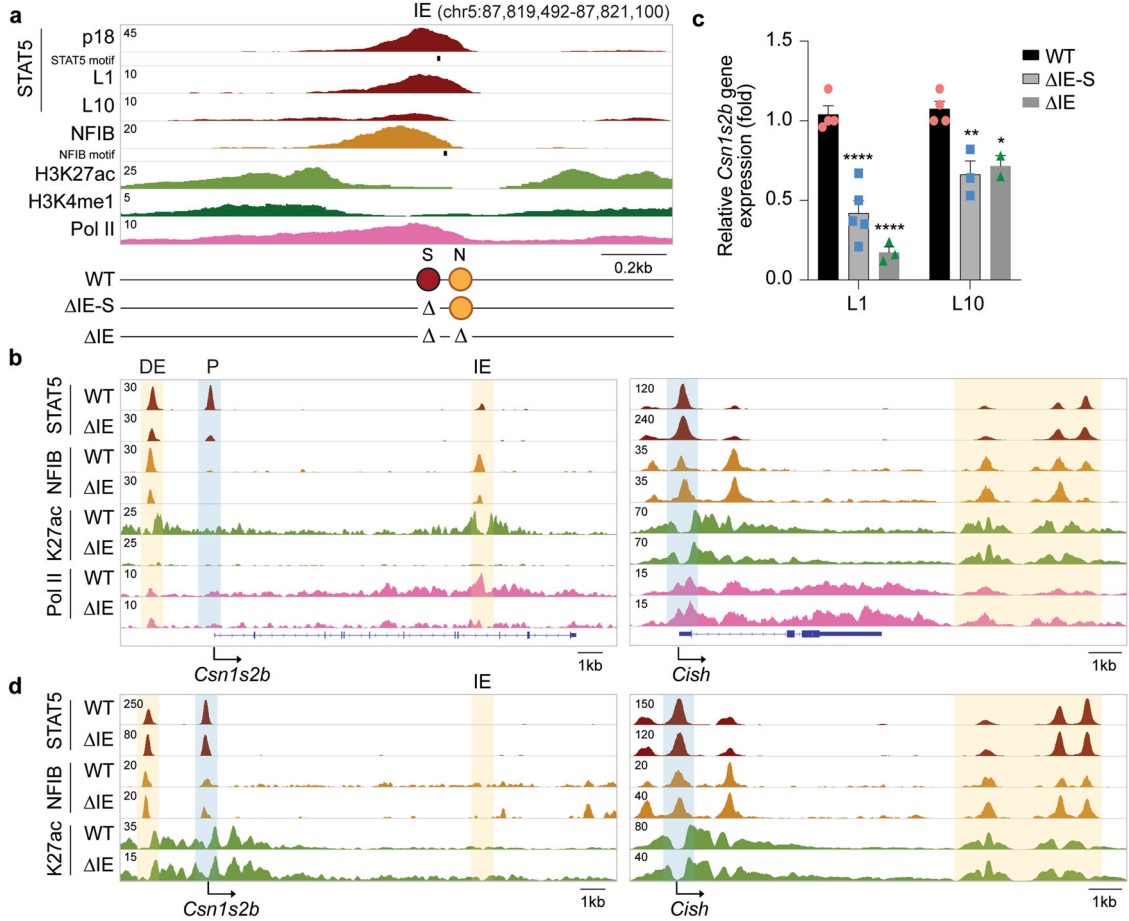

**Fig. 4 The intronic enhancer controls *Csn1s2b* expression in early lactation. a** Genomic feature of the *Csn1s2b* locus in mammary tissue at day one of lactation (L1) and diagram of mutant mice. ΔIE-S, deletion of the GAS motif; ΔIE, deletion of the entire intronic enhancer. **b** and **d** TF binding, H3K27ac marks and Pol II loading at the *Csn1s2b* locus in WT and mutant tissue at L1 (b) and L10 (d). The *Cish* locus served as a ChIP-seq control. **c** *Csn1s2b* mRNA levels were measured by qRT-PCR in L1 and L10 mammary tissue from WT and ΔIE mutant mice and normalized to *Gapdh* levels. Results are shown as the means ± s.e.m. of independent biological replicates (WT of L1 and L10, n = 4; L1 of ΔIE-S, n = 5; L10 of ΔIE-S, and L1 and L10 of ΔIE, n = 3). Two-way ANOVA followed by Tukey's multiple comparisons test was used to evaluate the statistical significance of differences. p-value = 0.0001, 0.0001, 0.0076 and 0.0478, respectively.

**Promoter activity is modulated by NFIB.** In addition to the distal and intronic enhancers, and the downstream SE, STAT5 and NFIB binding was also recorded in the promoter region within 100 bp of the TSS (Fig. 6a). We introduced an 18 bp deletion into the mouse genome leading to the loss of the NFIB site (Supplementary Table 3). *Csn1s2b* expression was reduced by ~60% (Fig. 6b), which coincided with reduced H3K27ac marks and Pol II coverage (Fig. 6c). Residual NFIB coverage could be the result of indirect binding through STAT5.

## Discussion

Here we use experimental mouse genetics and functionally investigate two mammary-gland enhancers and one super-enhancer that distinctly control expression of the *Csn1s2b* gene during pregnancy and lactation. A distal enhancer preferentially controls gene expression throughout lactation, an intronic enhancer is active in early lactation and a super-enhancer is needed for the activation of the locus during pregnancy (Fig. 7). We also gained new insight into the architecture and biology of redundant and non-redundant enhancer building blocks based on the mammary gland-enriched transcription factors STAT5 and NFIB.

Unlike most mammary gland enhancers, which are activated during early pregnancy and induce gene expression prior to

lactation[6,16], the *Csn1s2b* enhancer is preferentially established during lactation where it contributes to a several hundred fold expression induction. While most, if not all, mammary gland enhancers employ the cytokine-activated TF STAT5 as a core building block together with NFIB and GR, differential temporal recruitment has been observed. Although the underlying logic for a temporally distinct activation of seemingly identical enhancer sequences at different stages during mammary development, i.e. pregnancy versus lactation, is not clear that the respective TF binding sites might have different affinities. In support of this, genetic studies in mammary tissue have revealed that the concentration of STAT5 can influence gene activation patterns[6,17]. Alternatively, seemingly identical regulatory elements might become gradually accessible during differentiation, as shown in the α-globin locus[18]. The *Csn1s2b* distal enhancer is overtly more complex than other mammary enhancers[19] and contains two canonical and one non-canonical STAT5 binding sites in addition to an NFIB site and a GR half-site, which could contribute to its lactation-restricted activation.

Unlike other genetically validated enhancers where STAT5 binds to a canonical GAS motif[19–22], the *Csn1s2b* enhancer contains a functional non-canonical GAS motif in addition to two canonical sites. The contribution of canonical and non-canonical TF binding sites within enhancers is still being debated and it

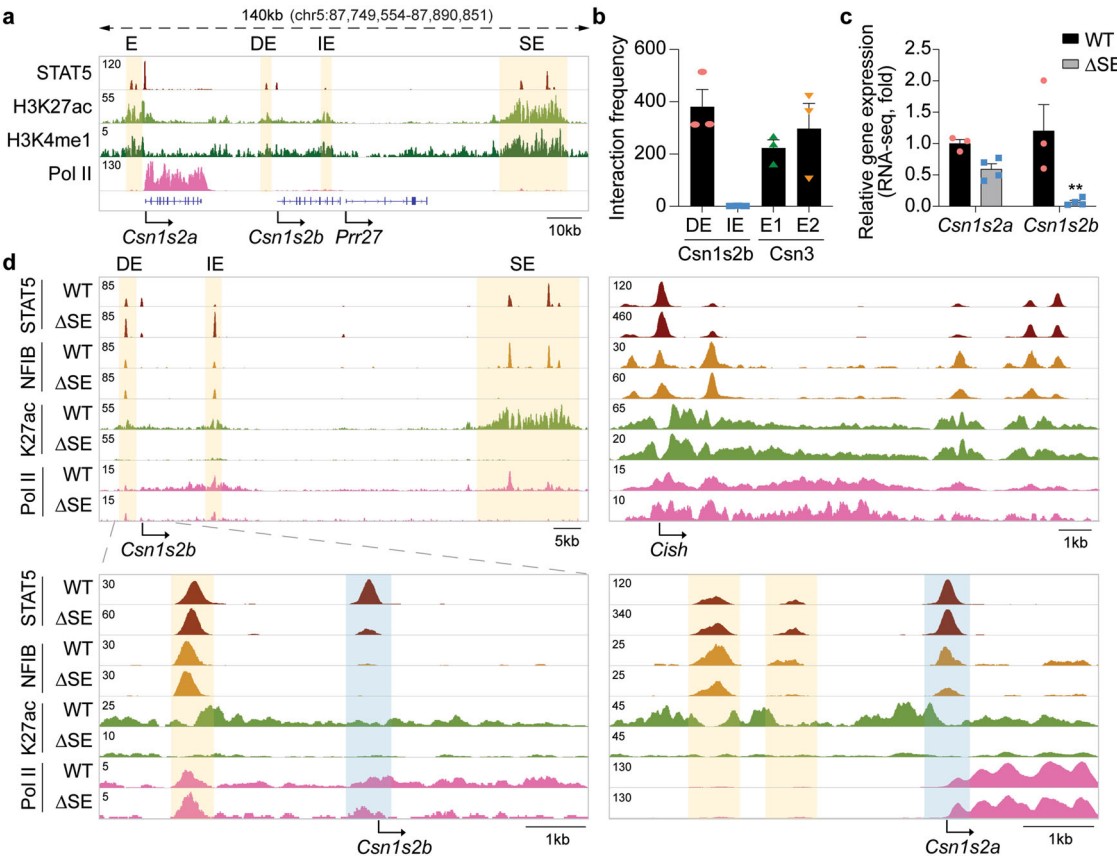

**Fig. 5 A super-enhancer activates Csn1s2b expression during pregnancy. a** Genomic feature of the *Csn1s2a/b* locus, including the downstream *Prr27* gene and the super-enhancer (SE) in L1 mammary tissue. **b** Interactions between SE and the regulatory elements of *Csn1s2b* and *Csn3* were detected by 3 C analysis. **c** *Csn1s2a/b* mRNA levels were measured by RNA-seq at day 18 of pregnancy (p18) in WT and ΔSE (deletion of the 10 kb SE) mice. Results are shown as the means ± s.e.m. of independent biological replicates (*n* = 4). Two-way ANOVA followed by Tukey's multiple comparisons test was used to evaluate the statistical significance of differences. Results are shown as the means ± s.e.m. of independent biological replicates (*n* = 3). *p*-value = 0.28 and 0.003 **d** STAT5 and NFIB binding, H3K27ac marks and Pol II coverage in ΔSE mammary tissue at L1. The *Cish* locus served as control.

might depend on the specific transcription factor and target tissue. STAT5 exists in two isoforms, STAT5A and STAT5B, which are encoded by two distinct genes[23]. In mammary tissue, STAT5A levels exceed STAT5B 2-3 fold[21] and STAT5A ChIP-seq experiments during pregnancy and lactation revealed that approximately 90% of all high-quality peaks coincide with the canonical GAS motif (TTCnnnGAA)[16]. Within mammary enhancers the percentage is even higher. It remains to be determined whether STAT5B recognizes the canonical GAS motif in mammary tissue. In liver STATB levels exceed STAT5A by approximately 10-fold and the majority of STAT5B ChIP-seq peaks coincide with the classical GAS motif[24]. A similar observation was made in T cells[20]. While GAS motifs with a 4 bp spacer are generally recognized by STAT6, another STAT member contributing to the differentiation of mammary alveolar cells[25], this non-canonical site is also recognized by STAT5 in the *Csn1s2b* distal enhancer.

Our finding that NFIB, a critical co-activator for a range of mammary genes, including *Csn1s2b*[26], can bind to the enhancer lacking the DNA binding motif adds further intrigue and suggests that the recruitment of multiple TFs can be facilitated through a single anchor, STAT5 in mammary enhancers. Our results also suggest that GR binding to its half-site within the distal enhancer might require the cooperative presence of a neighboring STAT5 site. Alternatively, GR could tether to STAT5 as shown in the *Wap* gene super-enhancer that contains a STAT5 binding motif but lacks a GR motif[19]. The progesterone receptor (PR)

binds to GR motifs[27] and is required for mammary alveolar development[28,29]. We analyzed PR ChIP-seq data from mammary tissue from progesterone treated non-parous mice[30] and no binding was detected at the *Csn1s2b* distal enhancer with a PR half-site motif. However, conclusions from these experiments are limited since they were conducted in non-parous mice that lack the differentiated alveolar compartment. Similarly, ChIP-seq data from the estrogen receptor (ER) did not reveal any binding[31]. It is conceivable that the presence of four TF binding motifs in the distal enhancer region permits additional TFs to bind through less conserved DNA binding sites. The presence of four TF binding blocks in the distal enhancer, possibly in synergy with additional promoter and intronic elements, enables high *Csn1s2b* expression levels during lactation. It remains to be elucidated why overtly equivalent enhancers activate other casein genes already during pregnancy. Of note, the more than 95% reduction of *Csn1s2b* expression caused by the deletion of the distal enhancer had no overt impact on lactation and is in agreement with other species that lack a functional *Csn1s2b* gene.

STAT5, GR and NFIB jointly occupy candidate enhancers of 'mammary-specific' genes[6,8,16] and their contributions in activating these genes during pregnancy and lactation have been investigated in mutant mice lacking these TFs. Since the global deletion of transcription factors can have widespread consequences on a given cell, such experiments may have limited impact on understanding their role on specific genes. As such, proliferation and differentiation of mammary epithelium during

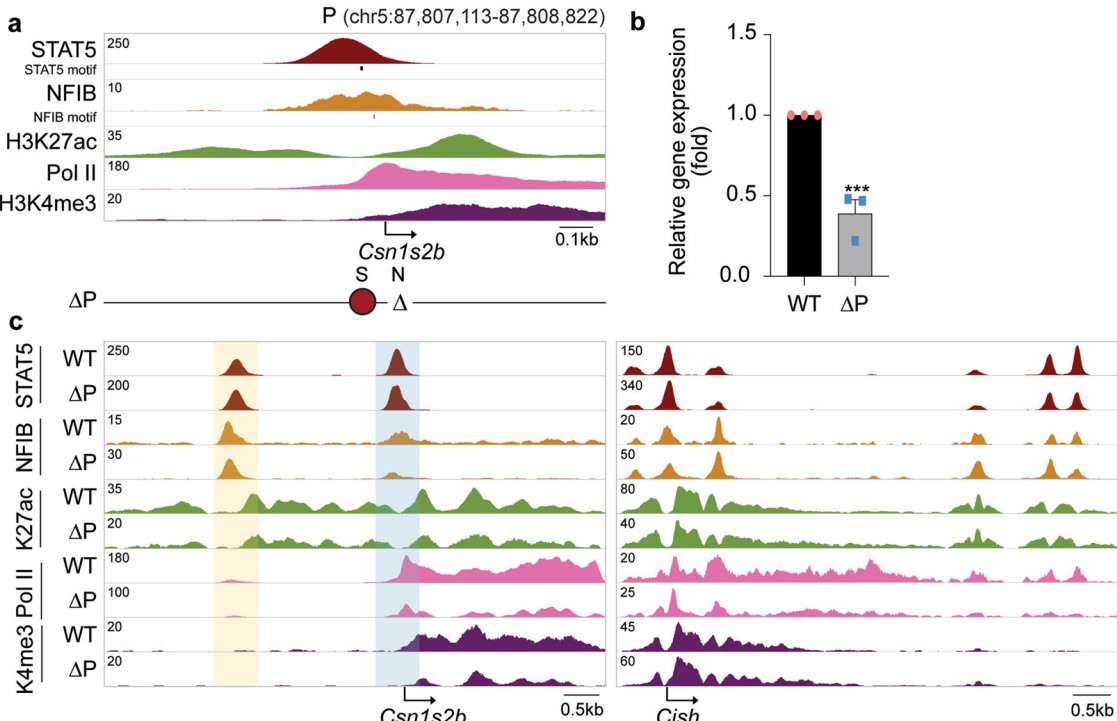

**Fig. 6 The NFIB site in the *Csn1s2b* promoter modulates gene activity. a** Genomic feature of the *Csn1s2b* promoter region in mammary tissue at day ten of lactation. ΔP, deletion of the NFIB motif. **b** mRNA levels of *Csn1s2b* gene were measured by qRT-PCR in L1 mammary tissue from WT and ΔP mutant mice and normalized to *Gapdh* levels. Results are shown as the means ± s.e.m. of independent biological replicates (*n* = 3). Unpaired two-tailed t-test was used to evaluate the statistical significance of differences. *p*-value = 0.002 **c** The consequences of super-enhancer deletion were confirmed by ChIP-seq analysis in WT and mutant tissues at L10. The *Cish* locus served as a ChIP-seq control.

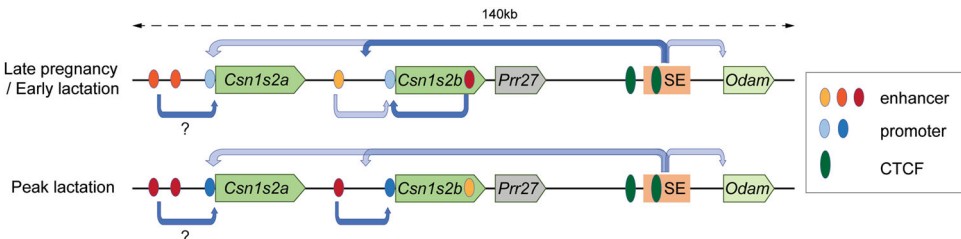

**Fig. 7 Proposed model of the activation of the *Csn1s2a/b* locus during pregnancy and lactation.** The *Csn1s2b* gene is under the control of three distinct enhancers; a distal enhancer controlling gene expression throughout lactation, an intronic enhancer active at late pregnancy and early lactation, and a super-enhancer (SE) required for the activation of the *Csn1s2b* locus. While the SE has limited influence on the expression of the *Csn1s2a* and *Odam* genes, it does not activate the *Prr27* gene in mammary tissue. Two candidate enhancers have been identified in the *Csn1s2a* distal region, but their function has not been validated through experimental genetics. The color intensity of promoter and enhancer elements at late pregnancy and peak lactation reflects their relative strengths. Darker colors indicate increased strength. Candidate *Csn1s2a* enhancer and promoter elements are shown but their biological strength is not known. CTCF binding sites have been identified[38].

pregnancy is greatly impaired in the combined absence of *Stat5a* and *Stat5b*, thus making it impossible to define their gene-specific roles. Targeted deletions of three STAT5 sites, in the mouse *Wap* locus, individually or in combination, defined redundant and non-redundant functions of an enhancer structure essential for gene activation during pregnancy[19]. In a classical study, Burdon and colleagues[32] employed transgenic mice to explore three STAT5 binding sites in a sheep β-lactoglobulin transgene[32] and suggested some degree of cooperativity between canonical and non-canonical STAT5 binding sites. However, not all genes under the control of STAT5 rely on cooperativity[20–22]. A role for the GR in mammary gene regulation is not that clear. Mice lacking the GR[33] or expressing a GR devoid of its DNA binding domain[34] displayed a slightly impaired development of mammary ducts in non-parous mice. However, alveolar differentiation during

pregnancy and expression of milk proteins during lactation were unimpaired, demonstrating that the GR is not required for normal mammary function and a potential compensatory function of the mineralocorticoid receptor has been proposed[33]. In contrast to the GR, the presence of the PR is required for the outgrowth and branching of mammary ducts[28,35]. However, since mice lacking the PR are infertile, mammary development and function during pregnancy has not been investigated. Moreover, PR ChIP-seq data from lactating mammary tissue are not available. However, studies using mouse mammary cell lines have revealed important functions of the PR in the regulation of the *Csn2* gene[36]. NFIB isoforms are abundantly expressed in mammary tissue[37] and deletion of the *Nfib* gene by itself or in combination with *Stat5* supported the concept of cooperativity in gene activation in mammary tissue[26].

In addition to the distal enhancer, we identified a super-enhancer (SE) essential for the activation of the *Csn1s2b*. We propose that this SE, which is located next to the *Odam* gene and separated from *Csn1s2b* by the *Prr27* gene, was part of the evolutionary older *Csn3* gene, and the younger *Csn1s2b* gene captured its activity. *Csn3* and its neighboring odontogenic ameloblast–associated (*Odam*) gene originated from a common precursor[9] and the gene arrangement in this locus (*Prr27*, *Odam*, *Fdcsp* and *Csn3*) predates the emergence of *Csn1s2b*. In contrast to *Csn1s2b*, pregnancy expression of the neighboring *Csn1s2a* gene is not overtly controlled by the SE. The molecular mechanism underlying lactation failure in mice lacking the SE needs further investigation and might be the result of deregulation of the entire locus. Two CTCF binding sites associated with this SE[38] might be an early signature of the casein locus before its expansion that added four additional casein genes, including *Csn1s2b*. However, deletion of these CTCF binding sites, did not alter *Csn1s2b* expression[38], suggesting that they have limited biological activity.

The increasing use of a wide range of ChIP-seq and chromatin capture approaches suggests that the mammalian genome is riddled with candidate enhancers that potentially control the spatio-temporal expression of lineage-specific genes[39–43]. However, as stated by one of the reviewers, we humans are not always good in selecting the TF sites or potential regulators that turn out to be important or essential. As shown here, uncovering the function and complexity of enhancers requires detailed genetic interventions. The casein locus with five genes expressed exclusively in mammary glands, three interspersed genes expressed in salivary glands and at least 20 candidate mammary enhancers, super-enhancers and CTCF sites remains a case study in evolutionary strategies to ensure uncompromised gene regulation. As further genetic inquiries are conducted, the multiplicity of regulatory building blocks controlling mammary- and salivary-gland specificity and cytokine-induced gene activation will continue to unfold.

## Methods

**Mice.** All animals were housed and handled according to the Guide for the Care and Use of Laboratory Animals (8th edition) and all animal experiments were approved by the Animal Care and Use Committee (ACUC) of National Institute of Diabetes and Digestive and Kidney Diseases (NIDDK, MD) and performed under the NIDDK animal protocol K089-LGP-17. CRISPR/Cas9 targeted mice were generated using C57BL/6 N mice (Charles River) by the transgenic core of the National Heart, Lung, and Blood Institute (NHLBI). Single-guide RNAs (sgRNA) were obtained from either OriGene (Rockville, MD) or Thermo Fisher Scientific (Supplementary Table 2). Target-specific sgRNAs and in vitro transcribed *Cas9* mRNA were co-microinjected into the cytoplasm of fertilized eggs for founder mouse production. The ΔN/S2 and ΔS2/3 mutant mouse was generated by injecting sgRNAs for NFIB site into zygotes collected from ΔS2 mutant mice. All mice were genotyped by PCR amplification and Sanger sequencing (Macrogen and Quintara Biosciences) with genomic DNA from mouse tails (Supplementary Table 3) and only homozygous mutant mice used in the study.

### Chromatin immunoprecipitation sequencing (ChIP-seq) and data analysis.
Mammary tissues from specific stages during pregnancy and lactation were harvested, and stored at −80 °C. The frozen-stored tissues were ground into powder in liquid nitrogen. Chromatin was fixed with formaldehyde (1% final concentration) for 15 min at room temperature, and then quenched with glycine (0.125 M final concentration). Samples were processed as previously described[21]. The following antibodies were used for ChIP-seq: STAT5A (Santa Cruz Biotechnology, sc-271542), GR (Thermo Fisher Scientific, PA1-511A), NFIB (Sigma-Aldrich, HPA003956), H3K27ac (Abcam, ab4729), RNA polymerase II (Abcam, ab5408), H3K4me1 (Active Motif, 39297) and H3K4me3 (Millipore, 07-473). Libraries for next-generation sequencing were prepared and sequenced with a HiSeq 2500 or 3000 instrument (Illumina).

The raw data were subjected to QC analyses using the FastQC tool (version 0.11.9) (https://www.bioinformatics.babraham.ac.uk/projects/fastqc/). Quality filtering and alignment of the raw reads was done using Trimmomatic[44] (version 0.36), Bowtie[45] (version 1.2.2) and Samtools[46] (version 1.8), with the parameter '-m 1' to keep only uniquely mapped reads, using the reference genome mm10. Picard

tools (version 2.9.2, Broad Institute. Picard, http://broadinstitute.github.io/picard/. 2016) was used to remove duplicates. Homer[47] (version 4.8.2) and DeepTools[48] (version 3.1.3) software was applied to generate bedGraph files, separately. Integrative Genomics Viewer[49] (version 2.5.3) was used for visualization. Each ChIP-seq experiment was conducted for more than two replicates. DeepTools was used to obtain the Pearson and Spearman correlation between the replicates.

In order to identify regions of ChIP-seq enrichment over the background, MACS[50] (version 2.2.7.1) peak finding algorithm was used. Peak calling of TFs and histone markers for WT and mutants was done for replicates, which were subsequently overlapped using Bedtools[51] (version 2.29.2) to identify high-confident peaks. TF bound enhancers were considered as true enhancer elements if they showed H3K27ac underneath. Coverage plots (normalized to 10 million reads) and motif analysis with default settings were done using Homer software.

Coverage plots were generated using Homer software with the bedGraph as input. R and the packages dplyr (https://CRAN.R-project.org/package=dplyr) and ggplot2[52] were used for visualization. Sequence read numbers were calculated using Samtools software with sorted bam files.

### RNA isolation and quantitative real-time PCR (qRT–PCR).
Total RNA was extracted from frozen mammary tissue of wild type and mutant mice using a homogenizer and the PureLink RNA Mini kit according to the manufacturer's instructions (Thermo Fisher Scientific). Total RNA (1 μg) was reverse transcribed for 50 min at 50 °C using 50 μM oligo dT and 2 μl of SuperScript III (Thermo Fisher Scientific) in a 20 μl reaction. Quantitative real-time PCR (qRT-PCR) was performed using TaqMan probes (*Csn1s2a*, Mm00839343_m1; *Csn1s2b*, Mm00839674_m1; mouse *Gapdh*, Mm99999915_g1, Thermo Fisher scientific) on the CFX384 Real-Time PCR Detection System (Bio-Rad) according to the manufacturer's instructions. PCR conditions were 95 °C for 30 s, 95 °C for 15 s, and 60 °C for 30 s for 40 cycles. All reactions were done in triplicate and normalized to the housekeeping gene *Gapdh*. Relative differences in PCR results were calculated using the comparative cycle threshold ($C_T$) method.

### Total RNA-seq analysis.
The frozen-stored tissues were ground into powder in liquid nitrogen and Total RNA was extracted using the PureLink RNA Mini kit according to the manufacturer's instructions (Thermo Fisher Scientific). Ribosomal RNA was removed from 1 μg of total RNAs and cDNA was synthesized using SuperScript III (Invitrogen). Libraries for sequencing were prepared according to the manufacturer's instructions with TruSeq Stranded Total RNA Library Prep Kit with Ribo-Zero Gold (Illumina, RS-122-2301) and 50 bp paired-end sequencing was done with a HiSeq 2500 instrument (Illumina).

The raw data were subjected to QC analyses using the FastQC tool (version 0.11.9) (https://www.bioinformatics.babraham.ac.uk/projects/fastqc/). Total RNA-seq read quality control was done using Trimmomatic[44] (version 0.36) and STAR RNA-seq[53] (version 2.5.4a) using 50 bp paired-end mode was used to align the reads (mm10). HTSeq[54] (version 0.9.1) was to retrieve the raw counts and subsequently, R (version 3.6.3) (https://www.R-project.org/), Bioconductor (version 3.10)[55] and DESeq2[52] were used. Additionally, the RUVSeq[56] package was applied to remove confounding factors. The data were pre-filtered keeping only those genes, which have at least ten reads in total. The visualization was done using dplyr (https://CRAN.R-project.org/package=dplyr) and ggplot2[57].

### Chromosome conformation capture (3 C).
DNA samples for 4C-seq from our previous study[58] were analyzed by qRT-PCR using SYBR green supermix (Biorad) on the CFX384 Real-Time PCR Detection System (Bio-Rad). The primers used were SE 5′-GTACTCTGGAAAAGTAGGCAGTGC-3′, Csn1s2b-DE 5′-AGCTGG CCAACACAAAAGAATGGC-3′, Csn1s2b-IE 5′- AGCCAGGTGAGTGAGCTAT GTTC-3′, Csn3-E1 5′- GAGTCTAACCACGCTACAGCTTC-3′, and Csn3-E2 5′-GTAGCTACTTCGGAAACCATCAAGG-3′. Interaction frequencies were normalized to the values of an internal control.

### Statistical analyses.
For comparison of samples, data were presented as standard deviation in each group and were evaluated with a one-way ANOVA followed by Dunnett's multiple comparisons test, 2-way ANOVA followed by Tukey's multiple comparisons test for comparisons or unpaired two-tailed t-test between WT and mutants using GraphPad Prism 8 (version 8.2.0). A value of *$P < 0.05$, **$P < 0.001$, ***$P < 0.0001$, ****$P < 0.00001$ was considered statistically significant. Significances for homer de novo motifs were evaluated with Poisson distribution[59].

**Reporting summary.** Further information on research design is available in the Nature Research Reporting Summary linked to this article.

## Data availability
The source data files were obtained or uploaded to Gene Expression Omnibus (GEO). ChIP-seq data of wild-type tissue at L1 and L10 were obtained under GSE74826[19], GSE115370[8], GSE145193[60], GSE127144 and GSE145193[60]. RNA-seq data for WT at p18, L1 and L10 were downloaded from GSE127140 and GSE115370[8]. The ChIP-seq and RNA-seq data from WT and mutant mice were uploaded in GSE161620. All files were summarized in Supplementary Data 1 and aligned to reference genome mm10.

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

## Acknowledgements

We thank Ilhan Akan, Sijung Yun and Harold Smith from the NIDDK genomics core for NGS. This work utilized the computational resources of the NIH HPC Biowulf cluster (http://hpc.nih.gov). This work was supported by the Intramural Research Programs (IRPs) of National Institute of Diabetes and Digestive and Kidney Diseases (NIDDK) and National Heart, Lung, and Blood Institute (NHLBI).

## Author contributions

H.K.L. and L.H. designed the study. C.L. generated mutant mice. H.K.L. and T.K. established mutant mouse lines. H.K.L. performed experiments and data analysis. H.K.L. and M.W. performed computational analysis. H.K.L. and L.H. supervised the study and wrote the manuscript. All authors approved the final version.

## Competing interests

The authors declare no competing interests.
