## [Peer Review File · Nature Communications]

REVIEWER COMMENTS

Reviewer #1 (Remarks to the Author):

This is an interesting manuscript that addresses the role, and the biological importance, of specific transcription factor (TF) binding sites in an enhancer element. The real value of this work is that functional studies of the TF binding motifs was carried out *in vivo* by precisely deleting sequences in the genome and then analysing gene expression in mice. The enhancer element chosen was a milk protein gene as milk protein genes are dramatically regulated during a pregnancy/lactation cycle. Interestingly, the *Csn1s2a* and *Csn1s2b* genes are differentially expressed with the latter being upregulated 1,000 fold between late pregnancy and lactation while *Csn1s2a* is expressed at approaching maximal levels already at p18.

The authors used ChIP- seq to identify a lactation-specific candidate enhancer. They then used CRISPR-Cas9 to delete selected TF binding motifs for NFIB and STAT5 and analysed gene expression. The data lead them to conclude that loss of specific TF sites leads to a shift of transcription factor binding to juxtaposed sites. The importance of this observation is that it suggests plasticity that is independent of direct protein-DNA interactions. This has significant implications for analysis of *in silico* studies.

This is a short manuscript but it represents a considerable amount of work that has been very carefully carried out by one of the leading laboratories in this field. I have no criticisms of the experimental work but have a few minor comments:

1. What are the consequences of deleting just the GAS motifs S1 and S2 or N and S1 and S2? Why were these specific combinations of TF motif deletions chosen?
2. The diagram in Figure 3a is confusing. Is a delta symbol missing for deletion of S3 motif?
3. In the Discussion, the authors may wish to refer to historic studies on Stat5 binding elements in other milk protein gene promoters and their interaction with each other and with additional transcription factors (BLG: Burdon, T et al., *Molecular Endocrinology* 1994 Nov;8(11):1528-36; WAP: Mukhopadhyay S.S et al., *Mol. Cell. Biol.* 2001 Oct;21(20):6859-69). In the BLG promoter, the weakest Stat5 binding site was suggested to only play a role in the presence of canonical binding sites, suggesting some degree of cooperativity between the three Stat5 binding sites.
4. It is of interest why there should be such a difference in transcriptional regulation. *Csn1s2a* is expressed at very high levels at p18 and does not increase much at L10 while *Csn1s2b* increases 1,000 fold between p18 and L10. Is this related to the different function of *Csn1s2b*, potentially in the transport of calcium phosphate in milk and would this be deleterious in pregnancy?
5. Could S3 bind Stat6? It is closer to a canonical Stat6 binding site than a Stat5 binding site (TTCnnnnGAA). Stat6 is known to be active during lactation and is important for mammary gland development during pregnancy.

Reviewer #2 (Remarks to the Author):

Review: Regulatory plasticity within a complex cytokine-sensing mammary gland enhancer

In this manuscript the authors describe the *in-vivo* functional analysis of putative regulatory elements involved in the spatial and temporal regulation of a mammary gland specific gene, *Csn1s2b*.

More specifically, they investigate how individual transcription factors and their binding sites contribute to this function and the regulation of gene expression. The authors report a number of CRISPR mediated deletion mutants affecting binding sites for STAT5 and NFIB and show that deletion mutants in certain sites have a more profound effect on the function of the regulatory element and gene expression than deletions of other sites.

Through analysis of chromatin organization and transcription factor binding they identified a number of potential regulatory regions close to and in the *Csn1s2b* gene binding STAT5, GR NF1B and MED1 and hypothesized that these would be important in the specific regulation of this gene.

STAT5 is an important player in the tissue specific regulation of gene expression in the mammary gland. However, it has long been established that it functions in concert with numerous other transcription factors (GR being one of the more prominent ones, as well as NFIB and others) all of which are needed for full gene expression capacity.

This is an ambitious and very involved study with a number of combinatorial transcription factor binding site deletions and analysis of their effect via ChIP-seq and RNA seq (although it is not totally clear what is exactly analyzed by RNA-seq and what by q-RT-PCR, this needs to be clarified). This study provides useful insights in the role of TF and regulatory elements besides the promoter in spatial and temporal gene regulation, although the overall concept is not new, and more temporal analysis would add significantly to the overall understanding of the functioning of these regulatory elements. Other studies in different tissues and species, some by these authors, have established that not all regulatory elements have profound effects (or the effects one anticipates) and that we humans are not always good in selecting the TF sites or potential regulators that turn out to be important or essential (at least in the context they are studied in). I bit more caution in the interpretation might be in place.

Some general issues:

Firstly, the title and premise of plasticity is not clear to me. Plasticity means adaptability to changes, and here the authors suggest plasticity in the site where a transcription factor binds. This is based on the fact that deletions of the full or part of a canonical binding site for STAT5 and NFIB have limited effects on *Csn1s2b* expression, while deletion of a non-canonical site has more profound effects.

There are several issues with this:

1) They take the viewpoint that only canonical binding sites could be of importance. However, it is well established that non-canonical or less optimal TF binding sites are important in regulatory elements often more so than "canonical" sites. I am sure upon more detailed analysis one could establish that many of the "non-canonical" sites are occupied by the transcription factor in-vivo.

2) They imply that STAT5 binding shifts from one site to another upon mutation of a site.

The ChIP-seq data has in no way high enough resolution to draw this conclusion.

As this is in part what underlies their interpretation of plasticity, I do not think that such an interpretation is warranted.

3) They do not assess or discuss if the 1 bp deletion in S2 is enough to affect possible binding of STAT5 at this site, nor do they address the fact that the delta-only deletion encompasses only one bp from the putative NFIB site and that all other deltaN deletions are different (varying from a nt change to full deletion) and how this could affect the results and the interpretation.

Secondly, the authors totally disregard the cooperative role of GR (and PR) in the functioning of STAT5 in mammary gland gene regulation. They show that GR is located at all enhancers they show and study (although, the IE is not shown and should be shown as well in Figure 1C). They omit any further mention of GR and do not incorporate the data they have for L1 and L10 in the WT from previous studies. On closer inspection there is a GR half site overlapping S1 in the Distal Enhancer which would account for the GR binding in the DE. There should be more attention paid to this and taken into account in their interpretation of the data as well as included in the discussion. The possible binding of other TFs that function in cooperation with each other and STAT5 (both in DE and IE) should also be addressed, as well as the possibility of the creation of new TF binding sites that could affect function or the enhancers studied.

Furthermore, the study of the intronic enhancer (IE) targets one GAS site, which has an effect on gene expression of *Csn1s2b* in lactation, but does not knock it out. Other TF binding sites like GR or additional STAT5 could be more important, furthermore STAT5 might not be so important as STAT5 binding seems to be lost in L10 or at least much lower than at P18 & L1. They seem to dismiss the importance of a limited effect (50% reduction at L1). As the author mention this could be a repressor element: does it have an effect on timing of expression? what happens at earlier time points? Is the induction from late pregnancy to lactation and from early lactation to L10 affected? Does it have an effect on flanking genes?

They argue that "this" is a very strong enhancer (I assume "this" is in regard to DE, please clarify), they omits the possible important cooperative effect of the IE which could as the authors mention have a repressive effect in pregnancy and early lactation, which is not very well explored

and could be an important component of the very strong and specific stage-specific expression regulation. Again, it is not explored or discussed if there is an effect at other time points e.g. at L1. What are the possible cooperative factors in IE?

Plasticity implies that the regulatory functions have activity regardless if a factor is bound on one or the other binding site or that one binding site can compensate for another. This is not what is shown in this study, the fact that a mutation of a canonical site has less, or no effect and deletion of a non-canonical site has, is not plasticity. Unless you can really show a difference in binding at the nt level between different mutants (based on base substitutions) and WT. I am not suggesting this should be done but only to tone down the interpretation of the data. Furthermore, I think that focusing on the "canonical" gas site might be a bit narrow. Maybe conclusion should be that canonical STAT5 site is not the most important binding element. That the cooperation is important, not sure if that entails plasticity.

Other comments:

--I suggest adding gland to mammary to title > mammary gland enhancer

--"Structural" studies is too ambiguous and vague, structural could also refer to the structure of the genome sequence such as indels, I suggest calling it chromatin structure, that is more appropriate in this context I believe.

---"we investigated a potential synergy between the prolactin-induced TF STAT5 and the mammary-enriched NFIB in the establishment of a lactation-specific enhancer that activates gene expression in mammary tissue several hundred-fold"

>How do you know that it is this enhancer that activates the expression several hundred-fold?

---"suggesting the presence of a unique enhancer that differentially respond to lactation hormones, with prolactin the most prominent one"

>Effects of inhibition of Prl signaling supports this (Ormandy 1997). Csn1s2b is most susceptible to Prl deletion or mutations.

--"STAT5 and NFIB binding coincided with their respective recognition motifs, suggesting their direct attachment to DNA."

>So does GR! Furthermore, the resolution of ChIP-seq data is not enough to discriminate between the different STAT5 sites that are so close"

--"These analyses exposed the plasticity within enhancers and the hitherto underappreciated concept that TF binding to their DNA motif is not required for their function."

--" Most intriguingly, there was no absolute requirement for some of the individual binding motifs as plasticity permitted binding of individual TF in the absence of their motif through juxtaposed components."

> See comments above regarding plasticity and TF-binding site use

--"Mechanistically, we propose that this unique activation is the result of STAT5 binding to a non-canonical site."

>what about other TF factors, and the cooperation of DE and IE and loss of binding of factors?

--"Our finding that NFIB, a critical co-activator for Csn1s2b, does not depend on its DNA recognition motif adds further intrigue and provides evidence that the recruitment of multiple TFs can be facilitated through a single anchor, STAT5 in mammary enhancers."

>see comments earlier about partial deletion of putative binding sites, and not addressing other factors that could be involved in recruiting.

Methods:

-CRISPR mutant generation:

Based on supplemental table 2 two sgRNA were used for the generation of deltaN, one of which is for the GAS motifs and identical as the one listed for the GAS motif (overlaps S1 and S2). Is this correct?? If so why were both used? Were both used for the generation of delta N/S2? What sgRNA was used for the generation of and delta S2/3 as the GAS gRNA sequence is specific for the S1 and S2 sequence?

Are the generated mutations present as homozygote or heterozygote in the animals?

-ChIP-seq:

The ChIP-seq enrichment analysis needs to be described in more detail so it can be reproduced by others.

-DATA availability:

It would be useful to provide a supplemental table listing in detail the data that were used in this study (antibody used, GSE number etc.) for existing data and the newly data generated in this study.

RNA-seq:

In complete and inaccurate descriptions of RNA-seq: missing details on isolation, library preparation, sequencing details (Single end vs paired, read length etc).

Data analysis description appears inaccurate and incomplete: Trimmomatic does not check for read quality, it trims reads (optionally based on read quality). Only using bowtie (and an old version of it) will not be appropriate for RNA-seq. List details (Software, version and parameter setting) how was RNA-seq data aligned and which genome assembly and annotation was used. How is differential expression analyzed?

Statistical analysis:

Needs more detail, why was ANOVA used was it one-way or two-way how was it performed, was any posthoc analysis performed (would have to be to get information on which groups were significantly different). ANOVA does not "ANOVA was used to evaluate the statistical significance of differences between WT and mutant mice" ANOVA determines if there is a difference between the means of the groups you are comparing, a post-hoc analysis is needed to determine between which samples the difference is significant.

Cannot really use normalized (set to 1) data in statistical analysis, what data was used to do the analysis?

Figure 1:

Call it Chromatin structure information instead of "structural information"

DE en IE are not indicated in fig1b

c-d: suggest to add the same information/figure for the EI region specially because that is studied here and not the Csn1s2a (could be moved to supplemental)

as mentioned it seems that GR/PR half sites are present at enhancers, please add

Suggest to use same color annotation throughout all figures: orange for enhancers and blue for promoters.

Figure 2: figure 2a indicate potential GR half site

figure 2b : please explain with more detail (in methods) how statistical analysis was performed for data (3b as well)

Fig2 c: why are different scales used for same ChIP-factor please sue same scale, or explain in legend. (fig3 c as well)

Please explain in more detail (in methods) how analysis (computational & statistical) was performed for figure 2. indicate n= doe each sample in legend text or figure (figure 3d as well)

Figure 3 see comments figure 2 b-d

Reviewer #3 (Remarks to the Author):

Lee and co-workers utilize their prior ChIP-seq data comparing chromatin marks and binding sites for the prolactin-activated STAT5, and other factors, to focus on regulatory regions for the *Csn1s2b* gene in mouse mammary tissue, in order to elucidate enhancer elements that regulate the strong induction of *Csn1s2b* during lactation (day L10). Crisper/Cas9 editing of the mouse genome is used to generate mice with deletions of putative regulatory sequences within a "distal" enhancer (3-4 kb upstream of *Csn1s2b*) at motifs for STAT5 and at motifs for NFIB binding, and at an intronic enhancer. The impact of these deletions on TF binding, chromatin marks, and gene expression at lactation day 10 is evaluated. Results indicate that deletion of STAT5 motif S3, which the authors describe as a non-canonical STAT5 site, when combined with deletion of the NFIB site or STAT5 motif S2, leads to >90% decrease in gene expression and decreased in factor binding. In contrast, deletion of the adjacent STAT5 motif S2 alone, or NFIB site alone, has little effect.

One major concern relates to limitations of the Crispr/Cas9 deletion data, and the author's interpretation that the STAT5 non-canonical site S3 has a unique critical role in gene induction. None of the constructs examine loss of site S3 alone. Furthermore, whereas the Δ S2 STAT5 motif mutation, which has little or perhaps no significant effect on gene expression, removes only a single nucleotide, and effectively mutates the motif from TTTNNGAA to ATTNNGAA, all three of the Δ S3 deletion mouse lines examined remove the entire S3 sequence plus what appear to be various lengths of flanking sequences (it is not exactly clear from table S3 which/how many of the adjacent nts are deleted). As a result, distances between the remaining STAT5 sites and the residual NFIB site are altered, which could contribute to the observed phenotype, rather than the loss of S3 per se. Construction of a sole Δ S3 site deletion with the same single nt deletion found in the Δ S2 site would seem to be required for a 'clean' experiment.

A second major concern is the narrow scope of the analysis, and the absence of follow up characterization and analysis of the most impactful deletions on accessibility (ATAC-seq) or interactions (4C-seq or Hi-seq). Further, the study is limited to regulatory regions in the immediate vicinity of *Csn1s2b*, as well as two regions at the adjacent *Csn1s2b* gene. As regulatory regions are often found even a few hundred kb from the TSS, the study could benefit greatly by analysis of the author's prior published data (as was done in Fig. 1) to identify other potential regulatory elements in the broader *Csn1s2* gene region that exhibit the same key properties of the *Csn1s2b* DE, namely, a strong increase in binding of STAT5 – or other TFs and marks – at lactation day L10, compared to p18 and L10.

Other specific comments

1. What is the physiological impact of the >90% decrease in *Csn1s2b* expression at L10? Is *Csn1s2b* protein or perhaps overall milk production decreased? is there any impact on pup weight gain? Are there any corresponding regulatory SNPs in the human gene ortholog?
2. Fig. 1c and 1d – Include data showing the differential binding of each factor at L10 vs. L10 and p18, as in Fig. 1b. also, include labels for DE and IE on the figure.
3. eRNA analysis – The method used – total DNA seq - - is quite inadequate, given its insensitivity for eRNA detection. Techniques such as GRO-seq or csRNA-seq are required.
4. Fig. 2d does not support the 40% decrease in H2K27ac at Δ S2 mentioned in the text. Also, in Fig. 2d, include error bard for WR, and include statistics. Same applies to Fig. 3d, and for WT bard in Fig. 2b, 3b, and 4b.
5. Fig. 2b - Conclusion of ns for Δ N/S2 is based on only n=3 mice. repeat PCR using a larger number of individual mice.
6. Discuss the significance of a non-canonical STAT5 site in the context of prior literature. Also, are there many other non-canonical STAT5 binding sites, e.g., in the author's mammary ChIP-seq data?
7. Provide genomic coordinates for all figures with genome browser screenshots, and include GEO accession numbers, with reviewer access, for all new high throughput RNA-seq and ChIP-seq dataset, etc.
8. State in methods whether the mouse studied are homozygous or heterozygous with respect to the deletions.

Response to reviewers

Reviewer #1 (Remarks to the Author):

This is an interesting manuscript that addresses the role, and the biological importance, of specific transcription factor (TF) binding sites in an enhancer element. The real value of this work is that functional studies of the TF binding motifs was carried out in vivo by precisely deleting sequences in the genome and then analyzing gene expression in mice. The enhancer element chosen was a milk protein gene as milk protein genes are dramatically regulated during a pregnancy/lactation cycle. Interestingly, the *Csn1s2a* and *Csn1s2b* genes are differentially expressed with the latter being upregulated 1,000-fold between late pregnancy and lactation while *Csn1s2a* is expressed at approaching maximal levels already at p18.

The authors used ChIP-seq to identify a lactation-specific candidate enhancer. They then used CRISPR-Cas9 to delete selected TF binding motifs for NFIB and STAT5 and analyzed gene expression. The data lead them to conclude that loss of specific TF sites leads to a shift of transcription factor binding to juxtaposed sites. The importance of this observation is that it suggests plasticity that is independent of direct protein-DNA interactions. This has significant implications for analysis of in silico studies.

This is a short manuscript, but it represents a considerable amount of work that has been very carefully carried out by one of the leading laboratories in this field. I have no criticisms of the experimental work but have a few minor comments:

1. What are the consequences of deleting just the GAS motifs S1 and S2 or N and S1 and S2? Why were these specific combinations of TF motif deletions chosen?

Response

We had planned to delete each of the GAS motifs, S1, S2, S3, and the NFIB site individually, but there were limitations for sgRNAs with high scores. The sgRNA we used initially in this study introduced the inactivating mutation at site S2 and these results are presented in Figure 2. Subsequently, we injected additional sgRNAs in WT and S2 mutant mice in order to target site N and we obtained founder mice carrying various patterns of deletions.

2. The diagram in Figure 3a is confusing. Is a delta symbol missing for deletion of S3 motif?

Response

We added the delta symbol for S3 deletion.

3. In the Discussion, the authors may wish to refer to historic studies on Stat5 binding elements in other milk protein gene promoters and their interaction with each other and with additional transcription factors (BLG: Burdon, T et al., *Molecular Endocrinology* 1994 Nov;8(11):1528-36; WAP: Mukhopadhyay S.S et al., *Mol. Cell. Biol.* 2001 Oct;21(20):6859-69). In the BLG promoter, the weakest Stat5 binding site was suggested to only play a role in the presence of canonical binding sites, suggesting some degree of cooperativity between the three Stat5 binding sites.

Response

We thank the reviewer for pointing out these classical and historical studies. The Burdon paper is based on transgenic mice and these results are pertinent to our work. We have now incorporated these studies in the discussion, distilling the knowledge of STAT5 and NFIB in five genes that have been investigated through mouse genetics (*Wap*, *Blg*, *IL2r* and *Csn1s2b*).

4. It is of interest why there should be such a difference in transcriptional regulation. *Csn1s2a* is expressed at very high levels at p18 and does not increase much at L10 while *Csn1s2b* increases 1,000-fold between p18 and L10. Is this related to the different function of *Csn1s2b*, potentially in the transport of calcium phosphate in milk and would this be deleterious in pregnancy?

Response

Csn1s2b is unique in that its activation occurs mainly during lactation, which is very distinct from other milk protein genes. The mechanism is not clear, and we have now discussed various possibilities, including temporal chromatin accessibility and transcription factor concentrations. It is not known whether *Csn1s2b* has a unique biological function as some mammalian species either have no *Csn1s2b* gene or a pseudogene.

The reviewer makes an interesting point that expression during pregnancy could have deleterious effects. To investigate this, it would be necessary to generate transgenic mice overexpressing *Csn1s2b* during pregnancy. This would go beyond the current study.

5. Could S3 bind Stat6? It is closer to a canonical Stat6 binding site than a Stat5 binding

site (TTCnnnnGAA). Stat6 is known to be active during lactation and is important for mammary gland development during pregnancy.

Response

Yes, absolutely, STAT6 could bind to S3, which as a 4 bp spacer in the GAS motif. We discuss this now and also the work by Watson and colleagues that have demonstrated a role for STAT6 in mammary development (PMID: 17611223). Unfortunately, there are no published STAT6 ChIP-seq data and the consequence of STAT6 deletion on *Csn1s2b* expression has not been published.

Reviewer #2 (Remarks to the Author):

In this manuscript the authors describe the in-vivo functional analysis of putative regulatory elements involved in the spatial and temporal regulation of a mammary gland specific gene, *Csn1s2b*.

More specifically, they investigate how individual transcription factors and their binding sites contribute to this function and the regulation of gene expression. The authors report a number of CRISPR mediated deletion mutants affecting binding sites for STAT5 and NFIB and show that deletion mutants in certain sites have a more profound effect on the function of the regulatory element and gene expression than deletions of other sites.

Through analysis of chromatin organization and transcription factor binding they identified a number of potential regulatory regions close to and in the *Csn1s2b* gene binding STAT5, GR NF1B and MED1 and hypothesized that these would be important in the specific regulation of this gene.

STAT5 is an important player in the tissue specific regulation of gene expression in the mammary gland. However, it has long been established that it functions in concert with numerous other transcription factors (GR being one of the more prominent ones, as well as NFIB and others) all of which are needed for full gene expression capacity.

This is an ambitious and very involved study with a number of combinatorial transcription factor binding site deletions and analysis of their effect via ChIP-seq and RNA-seq (although it is not totally clear what is exactly analyzed by RNA-seq and what by qRT-PCR, this needs to be clarified). This study provides useful insights in the role of TF and regulatory elements besides the promoter in spatial and temporal gene regulation, although the overall concept is not new, and more temporal analysis would

add significantly to the overall understanding of the functioning of these regulatory elements. Other studies in different tissues and species, some by these authors, have established that not all regulatory elements have profound effects (or the effects one anticipates) and that we humans are not always good in selecting the TF sites or potential regulators that turnout to be important or essential (at least in the context they are studied in). I bit more caution in the interpretation might be in place.

Response

We now clarified for each experiment the assays used to determine gene expression levels (qRT-PCR vs. RNA-seq). The reviewer was concerned about the temporal analyses. The *Csn1s2b* gene is pretty much silent throughout pregnancy and is activated at parturition and further throughout lactation. We therefore focused on the final day of pregnancy (day 18) and days 1 and 10 of lactation. The regulation of the *Csn1s2b* gene is accurately characterized by these developmental timepoints.

We have now included three additional mutant mouse lines (including one with a deletion of a super-enhancer that is located at 65 kb 3' of the *Csn1s2b* gene, that provide additional and novel information on the complexity of *Csn1s2b* regulation. As the reviewer states, "*we humans are not always good in selecting the TF sites or potential regulators that turnout to be important or essential*". With the reviewer's permission, we will use this statement in the discussion of our paper as it elegantly reflects the sentiment. Our study clearly supports this, and the *in vivo* deletion of candidate elements is required to determine their genuine function.

Although GR is a prominent TF binding to enhancers in 'mammary-specific' genes, its requirement for mammary development and milk protein gene expression is far from clear. Based on GR knock-out studies, mammary development during pregnancy appears to be rather normal (PMID: 12198239, 11581013, 15471946). We have now included a paragraph in the discussion that addresses the knowledge on GR and mammary gene expression *in vivo*, in mice.

Some general issues:

Firstly, the title and premise of plasticity is not clear to me. Plasticity means adaptability to changes, and here the authors suggest plasticity in the site where a transcription factor binds. This is based on the fact that deletions of the full or part of a canonical binding site for STAT5 and NFIB have limited effects on *Csn1s2b* expression, while deletion of a non-canonical site has more profound effects.

Response

The reviewer is correct, and we changed the title to “Redundant and non-redundant cytokine-activated enhancers control *Csn1s2b* expression in the mouse mammary gland during lactation”, which accurately reflects this study.

There are several issues with this:

1) They take the viewpoint that only canonical binding sites could be of importance. However, it is well established that non-canonical or less optimal TF binding sites are important in regulatory elements often more so than “canonical” sites. I am sure upon more detailed analysis one could establish that many of the “non-canonical” sites are occupied by the transcription factor in-vivo.

Response

Yes, the reviewer is absolutely correct that non-canonical sites and less optimal TF sites could be more important than canonical sites. For STAT5 the situation is likely somewhat different. We and others have extensively analyzed STAT5 ChIP-seq data over the past decade. While the vast majority of STAT5 peaks coincide with the classical GAS motif (TTCnnnGAA), we have not identified STAT5 binding to GAS motifs with a 2 bp spacer. The motif with a 4 bp spacer is recognized by STAT6 and possibly in some circumstances also by other STATs, such as STAT5 as shown in our study.

In mammary tissue, more than 90% of ChIP-seq STAT5 peaks coincide with the canonical GAS motif (PMID: 23324445). Specifically, almost all STAT5 ChIP-seq peaks at putative mammary enhancers (sites marked by H3K4me1 and H3K27ac) coincide with the canonical GAS motif. Over the past few years we have investigated the role of STAT sites in the regulation of three genes (*Wap*, *Stat5* itself and *Socs2*), and in all cases the relevant STAT5 peaks coincided with GAS motifs. STAT5 peaks can coincide with other TF motifs, such as NFIB, suggesting that they can piggy-back through other proteins. Yes, non-canonical binding sites are biologically significant and the degree to which this is relevant might depend on the transcription factor.

2) They imply that STAT5 binding shifts from one site to another upon mutation of a site. The ChIP-seq data has in no way high enough resolution to draw this conclusion. As this is in part what underlies their interpretation of plasticity, I do not think that such an interpretation is warranted.

Response

We agree that the ChIP-seq resolution does not allow us to make such a strong statement with certainty. We cautioned our interpretation as reflected in the revised title. But this does not alter our findings that the distal enhancer is composed of at least four TF binding sites with different significance.

3) They do not assess or discuss if the 1 bp deletion in S2 is enough to affect possible binding of STAT5 at this site, nor do they address the fact that the delta-only deletion

encompasses only one bp from the putative NFIB site and that all other deltaN deletions are different (varying from a nt change to full deletion) and how this could affect the results and the interpretation.

Response

The 1 bp deletion in S2 destroys the GAS motif (Δ S2) and results in reduced STAT5 binding to the distal enhancer as shown in Figure 2c. NFIB binding, H3K27c and Pol II loading are not significantly impaired in this mutant (Figure 2c) and *Csn1s2b* expression is also not significantly reduced (Figure 2b). These findings demonstrate that S2 is required for efficient STAT5 binding at the distal enhancer but that the transcriptional consequence seems to be minor.

We have clarified in the text the nature of the two different NFIB site deletions. The nature of the mutations used is fully transparent. The 15 bp Δ N deletion removes 1 bp from the NFIB site (TGGCA to TGGCT) and NFIB binding is not visibly impaired (Figure 2c). In contrast, although the NFIB site in the Δ N/S2 mutation is deleted, NFIB binding is still present (Figure 2c) suggesting that NFIB can bind to sequences coinciding with the GAS area. However, since STAT5 binding in this mutant is greatly curtailed, binding must occur through another, yet to be defined protein or through a site that does not resemble the TGGCA NFIB motif in any detectable way. This has now been discussed.

4) Secondly, the authors totally disregard the cooperative role of GR (and PR) in the functioning of STAT5 in mammary gland gene regulation. They show that GR is located at all enhancers they show and study (although, the IE is not shown and should be shown as well in Figure 1C). They omit any further mention of GR and do not incorporate the data they have for L1 and L10 in the WT from previous studies. On closer inspection there is a GR half site overlapping S1 in the Distal Enhancer which would account for the GR binding in the DE. There should be more attention paid to this and taken into account in their interpretation of the data as well as included in the discussion. The possible binding of other TFs that function in cooperation with each other and STAT5 (both in DE and IE) should also be addressed, as well as the possibility of the creation of new TF binding sites that could affect function, or the enhancers studied.

Response

We apologize to the reviewer that we did not include the vast knowledge of progesterone and glucocorticoids in mammary gland biology. This has now been rectified. We cited the literature that used gene knock-out mice and tissue culture cells. As the reviewer pointed out, GR binding at mammary enhancers as shown in several publications (e.g PMID: 27694626). However, the functional role of the GR in regulating mammary gene expression has not been established. Two studies using mice carrying a GR-null allele (PMID: 12198239) or a mutant GR lacking the DNA binding domain (PMID: 11581013) reported normal mammary development and gene expression, suggesting that the presence of the GR is not required for normal physiology. Having said this, it is possible that in the absence of GR other steroid receptors take its place.

As requested, we have now updated Figure 1c and show the area around the intronic enhancer. No GR binding has been detected during lactation, but very limited binding was detected at p18 (Supp Fig. 1b).

We conducted a de novo motif analysis of our GR ChIP-seq data (Supplementary Fig. 2), which did not reveal a classical consensus GR binding site. Instead, STAT5 (TTCnnnGAA) and NFIB (TGGCA/TGCCA) motifs were highly enriched. GR binds at the Csn1s2b enhancer but we were unable to identify a classical GR binding motif. It is possible that a less stringent motif cooperates with GAS and NFIB motifs. We pointed this out in the manuscript.

Regarding the PR, we included mouse and cell culture data from the Baylor groups in the discussion. Unfortunately, there are no PR ChIP-seq data from lactating mammary tissue which would help us to address the role of PR in the Csn1s2b gene.

5) Furthermore, the study of the intronic enhancer (IE) targets one GAS site, which has an effect on gene expression of Csn1s2b in lactation but does not knock it out. Other TF binding sites like GR or additional STAT5 could be more important, furthermore STAT5 might not be so important as STAT5 binding seems to be lost in L10 or at least much lower than at P18 & L1. They seem to dismiss the importance of a limited effect (50% reduction at L1). As the author mention this could be a repressor element: does it have an effect on timing of expression? what happens at earlier time points? Is the induction from late pregnancy to lactation and from early lactation to L10 affected? Does it have an effect on flanking genes?

Response

The reviewer is correct that besides that STAT5 sites, additional TF sites within the intronic enhancer (IE) might contribute to its function. Since this study was submitted in May, we have analyzed an additional mouse line in which we deleted 36 bp covering the GAS motif and the neighboring NFIB motif (Figure 4a). Mice from both lines were analyzed at L1 and L10 and a more profound reduction of Csn1s2b mRNA levels were detected in the larger deletion (Figure 4b), which coincided with reduced H3K27ac marks and also reduced TF occupancy at the promoter. These new data have been included and discussed.

6) They argue that “this” is a very strong enhancer (I assume “this” is in regard to DE, please clarify), they omits the possible important cooperative effect of the IE which could as the authors mention have a repressive effect in pregnancy and early lactation, which is not very well explored and could be an important component of the very strong and specific stage-specific expression regulation. Again, it is not explored or discussed

if there is an effect at other time points e.g. at L1. What are the possible cooperative factors in IE?

Response

We have added additional data about the consequences on the IE deletion on the expression of *Csn1s2b* and TF binding and histone marks at the distal enhancer and promoter (Figure 4). Specifically, we have included data from a new mouse line in which the entire IE core was deleted. These experiments have now been included and discussed. Yes, TF binding to the intronic enhancer is stronger during pregnancy than during lactation and could be a stage-specific repressor. But seems to be less likely, at least during late pregnancy, as no increased *Csn1s2b* mRNA levels are observed in mutant tissue at late pregnancy. Conducting further experiments during pregnancy would go beyond the current study and would not be possible at the current time.

7) Plasticity implies that the regulatory functions have activity regardless if a factor is bound on one or the other binding site or that one binding site can compensate for another. This is not what is shown in this study, the fact that a mutation of a canonical site has less, or no effect and deletion of a non-canonical site has, is not plasticity. Unless you can really show a difference in binding at the nt level between different mutants (based on base substitutions) and WT. I am not suggesting this should be done but only to tone down the interpretation of the data. Furthermore, I think that focusing on the “canonical” gas site might be a bit narrow. Maybe conclusion should be that canonical STAT5 site is not the most important binding element. That the cooperation is important, not sure if that entails plasticity.

Response

We fully agree with the reviewer and edited the manuscript accordingly.

Other comments:

8) I suggest adding gland to mammary to title
> mammary gland enhancer

Response

We changed the title to “Redundant and non-redundant cytokine-activated enhancers control *Csn1s2b* expression in the mouse mammary gland during lactation”

9) “Structural” studies is too ambiguous and vague, structural could also refer to the structure of the genome sequence such as indels, I suggest calling it chromatin structure, that is more appropriate in this context I believe.

Response

We changed “Structural” to chromatin structure.

10) “we investigated a potential synergy between the prolactin-induced TF STAT5 and the mammary-enriched NFIB in the establishment of a lactation-specific enhancer that activates gene expression in mammary tissue several hundred-fold”

> How do you know that it is this enhancer that activates the expression several hundred-fold?

Response

Our study was based on the hypothesis that the distal site bound by several TFs is the enhancer in question and this was tested. In our previous studies (PMID: 23275557, 30285185), we identified gene expression levels through RNA-seq. We summarized the induction of casein genes for pregnancy and lactation in Supplementary Table 1. Here, Csn1s2b is activated 358 folds between day6 of pregnancy (P6) and day 1 of lactation (L1) and 253 folds between L1 and L10. In addition, we observed that TF (STAT5, NFIB) occupancy at the distal enhancer site increased strongly at the pregnancy/lactation transition (between p18 and L1) (Figure 1).

11) “suggesting the presence of a unique enhancer that differentially respond to lactation hormones, with prolactin the most prominent one”

> Effects of inhibition of Prl signaling supports this (Ormandy 1997). Csn1s2b is most susceptible to Prl deletion or mutations.

Response

As the reviewer mentioned, Prl is a key component in mammary gland development for pregnancy and lactation. Here, we found Prl-sensing regulatory elements are differently activated and play a role in gene regulation for pregnancy.

12) “STAT5 and NFIB binding coincided with their respective recognition motifs, suggesting their direct attachment to DNA.”

> So does GR! Furthermore, the resolution of ChIP-seq data is not enough to discriminate between the different STAT5 sites that are so close”

Response

We agree that the close proximity of the GAS motif prevents a complete resolution of STAT5 binding. We identified the STAT5 and NFIB binding motifs under their ChIP-seq coverage and they are within the enhancer that is presented by the 'valleys' of activating histone (H3K27ac) marks. To clarify the roles of these binding motifs, we deleted sequences covering different TF motifs and conducted ChIP-seq for histone markers and TFs to determine the consequences of the mutations. We have now also included a de novo motif search using our GR ChIP-seq data (Supplementary Fig. 2).

13) "These analyses exposed the plasticity within enhancers and the hitherto underappreciated concept that TF binding to their DNA motif is not required for their function."

"Most intriguingly, there was no absolute requirement for some of the individual binding motifs as plasticity permitted binding of individual TF in the absence of their motif through juxtaposed components."

> See comments above regarding plasticity and TF-binding site use

Response

We agree and modified the manuscript accordingly.

14) "Mechanistically, we propose that this unique activation is the result of STAT5 binding to a non-canonical site."

> what about other TF factors, and the cooperation of DE and IE and loss of binding of factors?

Response

Among the TFs shown here to bind to mammary enhancers (STAT5, GR, NFIB), STAT5 appears to be the most important one. As stated earlier, deletion of GR from the mouse genome has little impact on mammary development and gene expression (PMID: 12198239) and also the deletion of NFIB (PMID: 24678731) has less consequences than that observed in the absence of STAT5 (PMID: 15340066).

Based on the new line of mice carrying a deletion inactivating the entire intronic, we have obtained further insight potential cooperation with the DE and this is being discussed.

Regarding other TF sites controlling Csn1s2b expression, we have identified a super-enhancer that is located at 65 kb 3' of the Csn1s2b gene. We have deleted this enhancer from the mouse genome and these data are shown in Figure 5.

15) "Our finding that NFIB, a critical co-activator for Csn1s2b, does not depend on its DNA recognition motif adds further intrigue and provides evidence that the recruitment

of multiple TFs can be facilitated through a single anchor, STAT5 in mammary enhancers.”

> see comments earlier about partial deletion of putative binding sites, and not addressing other factors that could be involved in recruiting.

Response

We couldn't find any putative GR binding sites including half site. Therefore, we couldn't say GR could bind directly on the enhancer.

Methods:

16) CRISPR mutant generation:

Based on supplemental table 2 two sgRNA were used for the generation of deltaN, one of which is for the GAS motifs and identical as the one listed for the GAS motif (overlaps S1 and S2). Is this correct?? If so why were both used? Were both used for the generation of delta N/S2? What sgRNA was used for the generation of and delta S2/3 as the GAS gRNA sequence is specific for the S1 and S2 sequence?

Are the generated mutations present as homozygote or heterozygote in the animals?

Response

We corrected sgRNA sequence and one sgRNA used to introduce ΔN . $\Delta N/S2$ was generated by injecting the sgRNA for NFIB site in the $\Delta S2$ mice. All mutant mice were established homozygous lines and used to investigate biological consequence.

17) ChIP-seq:

The ChIP-seq enrichment analysis needs to be described in more detail so it can be reproduced by others.

Response

We added the pipeline for the ChIP-seq enrichment analysis in the M&M part. In order to identify regions of ChIP-seq enrichment over the background, we used MACS (version 2.2.7.1) peak finding algorithm. Peak calling of TFs and histone markers for WT and mutants was done for replicates, which were subsequently overlapped using Bedtools (version 2.29.2) to identify high-confident peaks. TF bound enhancers were considered as true enhancer elements if they showed H3K27ac underneath. Coverage plots (normalized to 10 million reads) and motif analysis with default settings were done using Homer software. Coverage plots were generated using Homer software with the bedGraph as input. R and the packages dplyr and ggplot2 were used for visualization.

DeepTools was used to obtain the Spearman correlation between the replicates and the results were summarized in Supplementary Table 4.

18) DATA availability:

It would be useful to provide a supplemental table listing in detail the data that were used in this study (antibody used, GSE number etc.) for existing data and the newly data generated in this study.

Response

We summarized all NGS data, including public data and new dataset, used in this paper and their GEO numbers in Supplementary Table 4.

19) RNA-seq:

In complete and inaccurate descriptions of RNA-seq: missing details on isolation, library preparation, sequencing details (Single end vs paired, read length etc). Data analysis description appears inaccurate and incomplete: Trimmomatic does not check for read quality, it trims reads (optionally based on read quality). Only using bowtie (and an old version of it) will not be appropriate for RNA-seq. List details (Software, version and parameter setting) how was RNA-seq data aligned and which genome assembly and annotation was used. How is differential expression analyzed?

Response

We apologize for the oversight and we have now added all information for RNA isolation, library preparation, software and version in the pipeline and additional analyses in the Method section.

20) Statistical analysis:

Needs more detail, why was ANOVA used was it one-way or two-way how was it performed, was any post hoc analysis performed (would have to be to get information on which groups were significantly different). ANOVA does not “ANOVA was used to evaluate the statistical significance of differences between WT and mutant mice” ANOVA determines if there is a difference between the means of the groups you are comparing, a post-hoc analysis is needed to determine between which samples the difference is significant. Cannot really use normalized (set to 1) data in statistical analysis, what data was used to do the analysis?

Response

We performed post hoc analysis and added detailed information for statistics in each figure in Method and Figure legends. This is stated in the manuscript.

21) Figure 1:

Call it Chromatin structure information instead of “structural information”
DE and IE are not indicated in fig1b

Response

We changed the word as the reviewer’s suggestion and added the indication for those enhancers in Figure 1.

22) c-d: suggest to add the same information/figure for the EI region specially because that is studied here and not the Csn1s2a (could be moved to supplemental) as mentioned it seems that GR/PR half sites are present at enhancers, please add
Suggest to use same color annotation throughout all figures: orange for enhancers and blue for promoters.

Response

We changed the figure to include the IE region, but kept the figure 1c for the Csn1s2a locus to show evolution of Csn1s2 genes and difference between Csn1s2a and Csn1s2b genes. We used the unified color codes for enhancers and promoters in all figures.

23) Figure 2: figure 2a indicate potential GR half site

Response

We checked enhancer sequences for GR half sites or any other known sites. However, we couldn’t find any putative GR sites (RGXACAnnnTGTXCY). To obtain more information about putative GR binding sites in lactating mammary gland tissue, we conducted a de novo motif search using our GR ChIP-seq data (Supplementary Fig. 2).

24) figure 2b: please explain with more detail (in methods) how statistical analysis was performed for data (3b as well)

Response

We added the information for statistical analysis in the figure legend.

25) Fig2 c: why are different scales used for same ChIP-factor please use same scale, or explain in legend. (fig3 c as well)

Response

Although we tried to get similar number of reads in ChIP-seq, it is impossible to get the same number of reads and the same peak heights, especially on the same sites. So, we added control loci to show ChIP-seq quality and used the ratio between data into the Csn1s2b locus.

26) Please explain in more detail (in methods) how analysis (computational & statistical) was performed for figure 2. indicate n= doe each sample in legend text or figure (figure 3d as well)

Response

We added detailed information for number of each samples and statistical analysis in the figure legends.

27) Figure 3 see comments figure 2 b-d

Response

We added same edits in Figure 3.

Reviewer #3 (Remarks to the Author):

Lee and co-workers utilize their prior ChIP-seq data comparing chromatin marks and binding sites for the prolactin-activated STAT5, and other factors, to focus on regulatory regions for the Csn1s2b gene in mouse mammary tissue, in order to elucidate enhancer elements that regulate the strong induction of Csn1s2b during lactation (day L10). Crisper/Cas9 editing of the mouse genome is used to generate mice with deletions of putative regulatory sequences within a “distal” enhancer (3-4 kb upstream of Csn1s2b) at motifs for STAT5 and at motifs for NFIB binding, and at an intronic enhancer. The impact of these deletions on TF binding, chromatin marks, and gene expression at lactation day 10 is evaluated. Results indicate that deletion of STAT5 motif S3, which the authors describe as a non-canonical STAT5 site, when combined with deletion of the NFIB site or STAT5 motif S2, leads to >90% decrease in gene expression and

decreased in factor binding. In contrast, deletion of the adjacent STAT5 motif S2 alone, or NFIB site alone, has little effect.

One major concern relates to limitations of the Crispr/Cas9 deletion data, and the author's interpretation that the STAT5 non-canonical site S3 has a unique critical role in gene induction. None of the constructs examine loss of site S3 alone. Furthermore, whereas the Δ S2 STAT5 motif mutation, which has little or perhaps no significant effect on gene expression, removes only a single nucleotide, and effectively mutates the motif from TTTNNNGAA to ATTNNGAA, all three of the Δ S3 deletion mouse lines examined remove the entire S3 sequence plus what appear to be various lengths of flanking sequences (it is not exactly clear from table S3 which/how many of the adjacent nts are deleted). As a result, distances between the remaining STAT5 sites and the residual NFIB site are altered, which could contribute to the observed phenotype, rather than the loss of S3 per se. Construction of a sole Δ S3 site deletion with the same single nt deletion found in the Δ S2 site would seem to be required for a 'clean' experiment.

Response

We thank the reviewer for the constructive comments. Yes, the Δ S2 deleted 1 bp from the canonical GAS motif (TTCctgGAA to TCctgGAA). Based on all published information about STAT binding sites, this mutation should abrogate STAT5 binding and this is supported by Figure 2c. The residual STAT5 binding in this mutation can be explained by the presence of S2 and S3. In a previous study from our lab (PMID: 27376239), we mutated one or two nucleotides within GAS motifs and NFIB and ELF5 binding sites in the Wap gene and this resulted in the inability of enhancers to form. Also, other reference for IL2r enhancers showed the same findings. We clarified the exact positions and sizes of the different deletions for each mutant line in Supplementary Table 3.

Yes, the reviewer is correct that a complete molecular understanding of the enhancer beyond a shadow of doubt would require the individual deletion of S3. At this point we have not succeeded in obtaining such a specific deletion. As the reviewer knows, and we have published on this issue (PMID: 28561021), CRISPR/CAS9 has limitations. Also, deaminase base editing at this site is problematic due to the absence of sgRNAs with a high score. Lastly, given the current situation, it is not possible to generate this mutation.

A second major concern is the narrow scope of the analysis, and the absence of follow up characterization and analysis of the most impactful deletions on accessibility (ATAC-seq) or interactions (4C-seq or Hi-seq). Further, the study is limited to regulatory regions in the immediate vicinity of Csn1s2b, as well as two regions at the adjacent Csn1s2b gene. As regulatory regions are often found even a few hundred kb from the TSS, the

study could benefit greatly by analysis of the author's prior published data (as was done in Fig. 1) to identify other potential regulatory elements in the broader Csn1s2 gene region that exhibit the same key properties of the Csn1s2b DE, namely, a strong increase in binding of STAT5 – or other TFs and marks – at lactation day L10, compared to p18 and L10.

Response

The reviewer is correct in that additional regulatory elements outside the immediate Csn1s2b locus might be relevant. We have identified a super-enhancer (SE) 65 kb 3' of the Csn1s2b gene. Notably another gene, Prr27, is located between the SE and the Csn1s2b gene. We had generated mice with a 10 kb deletion spanning this SE and we have analyzed these mutants after submission of our manuscript in May. We have now included data that this SE controls Csn1s2b expression and ChIP-seq and 3C data that reveal the chromatin structure and TF binding and demonstrate that the SE effectively communicates with the distal enhancer and promoter. We have integrated these data with our prior studies on

We had also generated a mouse line carrying a deletion in the NFIB promoter site. This deletion had statistically insignificant consequences, both on the level of gene expression and activating histone marks.

Other specific comments

1. What is the physiological impact of the >90% decrease in Csn1s2b expression at L10? Is Csn1s2b protein or perhaps overall milk production decreased? is there any impact on pup weight gain? Are there any corresponding regulatory SNPs in the human gene ortholog?

Response

Lactation, as measured by the growth of pups, was not affected in mice producing only about 2% of Csn1s2b (deletion of the distal enhancer). Csn1s2b is less abundant than other caseins suggesting that it is not needed for lactation and the nutrition of the pups. In contrast, publications have shown that Csn2 and Csn1s2a are required for pup growth and survival. We have discussed this in the manuscript. Other species, such as humans, carry a non-functional Csn1s2b gene supporting the notion that this milk protein is not required for successful lactation.

2. Fig. 1c and 1d – Include data showing the differential binding of each factor at L10 vs. L10 and p18, as in Fig. 1b. also, include labels for DE and IE on the figure.

Response

Figure 1b provides information on the temporal induction of TF binding and histone modifications at candidate enhancers associated with the two Csn1s2 genes. Figures 1c and d present chromatin structure for each of the two Csn1s2 genes at day 10 of lactation (L10). We have added additional data in Supplementary Figure 1.

3. eRNA analysis – The method used – total DNA seq - - is quite inadequate, given its insensitivity for eRNA detection. Techniques such as GRO-seq or csRNA-seq are required.

Response

The reviewer is correct that a full picture of eRNAs at the various enhancers would require techniques other than total RNA-seq. We have therefore deleted this part from the text. Having said this, the mere fact that we have identified a clear eRNA signature at the intronic enhancer suggests that it is highly transcribed, probably more so than the distal enhancer.

4. Fig. 2d does not support the 40% decrease in H2K27ac at Δ S2 mentioned in the text. Also, in Fig. 2d, include error bars for WT, and include statistics. Same applies to Fig. 3d, and for WT bars in Fig. 2b, 3b, and 4b.

Response

We appreciated the reviewer pointing out the data for Δ S2. We found the H3K27ac activity of Δ N/S2 decreased ~ 40% similar to gene expression data. We also added error bars for WT and statistics between WT and mutant groups.

5. Fig. 2b - Conclusion of ns for Δ N/S2 is based on only n=3 mice. repeat PCR using a larger number of individual mice.

Response

We repeated qRT-PCR with original three and additional three samples for Δ N/S2 and added the conclusion from them in the figure.

6. Discuss the significance of a non-canonical STAT5 site in the context of prior literature. Also, are there many other non-canonical STAT5 binding sites, e.g., in the author's mammary CHIP-seq data?

Response

Over the years we have conducted a large number (100s) of STAT5 ChIP-seq data from different stages of mammary gland development, throughout pregnancy, lactation and involution from wt mice and various engineered mutants. In general, more than 90% of the STAT5 peaks coincide with the canonical GAS motif (TTCnnnGAA). We have never observed a motif with a 2 bp spacer. In genes with mammary enhancers, approximately 95% of STAT5 peaks are associated with the canonical GAS motif. In cases where no GAS motif is found, we frequently identify an NFIB motif (TGGCA/TGCCA) suggesting that STAT5 binds through NFIB at these sites. We have discussed this in the manuscript.

It has been reported that STAT5 binds to the STAT consensus binding site TTCNNNGAA (N3) with high affinity and is able to bind on TTCNNNNGAA (N4) with low affinity (PMID: 15677477, 11053426, 9852045, 11850439).

7. Provide genomic coordinates for all figures with genome browser screenshots, and include GEO accession numbers, with reviewer access, for all new high throughput RNA-seq and ChIP-seq dataset, etc.

Response

We added a list for all NGS data and their GEO numbers, including new dataset, in Supplementary Table 4.

8. State in methods whether the mouse studied are homozygous or heterozygous with respect to the deletions.

Response

We have used homozygous mice in all experiments and clarified it in method part.

REVIEWER COMMENTS

Reviewer #2 (Remarks to the Author):

Review revised manuscript

I appreciate the extent to which the authors have addresses reviewers' comments. The additional new data included in the manuscript provide a somewhat more complete picture of the complex regulation of this gene cluster. Although it is a bit fraud probably to discuss the *csn1s2b* gene regulation without the context of the whole cluster, even though it appears to be regulated in a unique manner. As is illustrated by the mention--as a sidebar--that the SE interacts with upstream regulatory regions close to *Csn3*. It is also clear that with the deletion of the SE the whole chromatin landscape appears to have been affected (no *k27Ac* in the parts that are shown)

Adding this bit of information on the Super enhancer (SE) begs the question: what are the effect on expression of other genes in the regions, this might be an important regulator for the whole gene cluster. The authors only mention effect at *p18* on *Csn1s2b* (which is expressed at very low levels at this time) and *Csn1s2b*, and the failure to lactate. Is this failure because the chromatin in the whole gene cluster is not organized properly? Affecting expression of many of the genes in this region?

You are welcome to quote me on the limitations of humans (or biases we have) in picking and prioritizing factors important in gene regulation.

I still have some issue with how the different deletion variants for the NF1A binding site are discussed in the manuscript: firstly, if I understand it correctly NF1 DBM is palindromic (as so many TF binding site) and only a half-site can be identified in the DE; Secondly, not all Delta N are the same, some fully delete other result in base changes and based on the NF1B consensus site (CTTGGCANNNTGCCAA) and logo sequence it appears that only dN-S2 and dN-S3 would really delete the site or make it dysfunctional, the others could be predicted to be still functional.

Regarding the GR sites. I appreciate that the authors have addressed GR in their analysis now. However, they fail to address the occurrence of Nuclear receptor half-sites that are probably important in cooperative binding, specific gene regulation and GR's tissue specific function (doi: 10.1101/gr.188581.114, doi: 10.1186/s13059-014-0418-y) and the similarity of AR and GR (half-) sites (DOI 10.1007/s00018-017-2467-3) of which they identify a good number in the GR binding site Motif analysis. As it is an AR-half site motif that overlaps the S1 STAT binding motif (see below) and this is affected or deleted in dN-S3 and dN-S1/3. I do not think it changes the overall conclusions of the manuscript but for completeness it should be addresses in a bit more detail.

see attached document)

NF1B STAT5 GR-half-site

```
WT TAAAGAGATGGCAAGTGAGCTCAGGGCTCT...ATGTTCTCTGAATCTATTCTGGAAAAG
dS2 TAAAGAGATGGCAAGTGAGCTCAGGGCTCT...ATGTTCTCTGAATCTAT-CCTGGAAAAG
dN* TAAAGAGATGGC-----TCT...ATGTTCTCTGAATCTATTCTGGAAAAG
dNS2 TAAAGA-----TCAGGGCTCT...ATGTTCTCTGAATCTAT-CCTGGAAAAG
WT NNNNNAGATGGCAAGTGA...TTCTTCTGTTGAAATC...ATGTTCTCTGAATCTATTCTGGAAAAG
dNS3 dG NNNNNAGATGTTC-----TTCTTCTGAATCTATTCTGGAAAAG
dN*S13 dG NNNNNAGATGGCCA-----TCTATTCTGGAAAAG
dS2/3 NNNNNAGATGGCAAGT-----ATGTTCTCTGAATCTAT-CCTGGAAAAG
```

(see attachment) logo motifs for

NF1B

STAT5

AR-half site

GR full site

Some minor points

Line 59 & 87: size/ definition of extend of casein locus ~400kb or ~280 kb?

Line273: "...where it induces gene expression several hundred-fold..." Suggest: ..where it contributes (enables) to a several hundred fold expression induction...

Line 275/276: appears to be an incomplete sentence

Line 305: "...absence of an overt biding site.." see above: there appears to be an AR (GR) half-site.

Line: 308: conceptually, I take issue calling it "strength" of an enhancer, IMO the enhancer enables/mediates (possibly in conjunction with the EI in pregnancy) the exceptionally high transcription of the csn1s2b gene to only initiate/occur during lactation while the other genes in this gene cluster initiate such high transcription at an earlier point (pregnancy-early lactation)

Line321: "...Three STAT binding in..." seems incomplete: binding sites?

Discussion regarding GR and PR: there is virgin PR Chip-seq data and I believe there is extensive overlap with GR bound sites in lactation. (PR repression might be another part of the puzzle)

R#3 from the previous round

(1) #1 - concerning *physiol.* Significance of a >90% decrease in (*sn152b*): please summarise and incorporate your response in the manuscript text.

Response

We included this information (lines 314-319).

(2) Specific comment #7 - Genomic coordinates should be shown on all genome browser screenshots.

Response

We edited the figures accordingly.

Response to reviewer #2

*I appreciate the extent to which the authors have addresses reviewers' comments. The additional new data included in the manuscript provide a somewhat more complete picture of the complex regulation of this gene cluster. Although it is a bit fraud probably to discuss the *csn1s2b* gene regulation without the context of the whole cluster, even though it appears to be regulated in a unique manner. As is illustrated by the mention--as a sidebar--that the SE interacts with upstream regulatory regions close to *Csn3*. It is also clear that with the deletion of the SE the whole chromatin landscape appears to have been affected (no k27Ac in the parts that are shown).*

*Adding this bit of information on the Super enhancer (SE) begs the question: what are the effect on expression of other genes in the regions, this might be an important regulator for the whole gene cluster. The authors only mention effect at p18 on *Csn1s2b* (which is expressed at very low levels at this time) and *Csn1s2b*, and the failure to lactate. Is this failure because the chromatin in the whole gene cluster is not organized properly? Affecting expression of many of the genes in this region?*

Response

We thank this reviewer for the positive comments. Yes, the super-enhancer possibly plays a significant role in controlling the regulation of this eight-gene locus in mammary tissue and maybe also in other tissues, such as salivary glands. The locus under investigation is composed of at least eight genes, the five casein genes that are expressed exclusively in mammary tissue and three genes (*Prr27*, *Odam* and *Fdcsp*), which are highly expressed in salivary tissue and possibly to some extent in dental tissue. While the current study focuses on the regulation of the uniquely expressed *Csn1s2b* gene, additional investigations are underway to determine the biology of this super-enhancer in the context of enhancers associated with the other four casein and the three salivary genes. As shown in this paper, *Csn1s2a*, the twin of *Csn1s2b*, appears not to be under the control of the super-enhancer. We added additional information in the discussion section (lines 354-357).

I still have some issue with how the different deletion variants for the NF1A binding site are discussed in the manuscript: firstly, if I understand it correctly NF1 DBM is palindromic (as so many TF binding site) and only a half-site can be identified in the DE; Secondly, not all Delta N are the same, some fully delete other result in base changes and based on the NFIB consensus site (CTTGGCANNNTGCCAA) and logo sequence it appears that only dN-S2 and dN-S3 would really delete the site or make it dysfunctional, the others could be predicted to be still functional.

Response

Yes, the original nuclear factor 1 (NF1) sites (bound by NFIA, NFIB, NFIB and NFIX) were characterized by the palindromic motif TGGCAnnnTGCCA¹⁻³. However, additional research has shown that the vast majority of NF1 binding sites is characterized by half-site motifs. In mammary tissue, NFIB is the most abundant member of this family but its presence is not required for normal mammary development⁴.

Yes, the NFIB half-site motif is completely deleted in Δ N-S2 and Δ N-S3. However, mutations Δ N and Δ N/S1/3 disrupt the canonical NFIB half-site motif and based on current knowledge this should disrupt TF binding.

Although mutation Δ S2/3 retains the wt NFIB half-site motif, NFIB binding is completely lost, demonstrating the physical presence of a binding site by itself is not sufficient for TF binding. We added additional information in the text (lines 182-184).

Regarding the GR sites. I appreciate that the authors have addressed GR in their analysis now. However, they fail to address the occurrence of Nuclear receptor half-sites that are probably important in cooperative binding, specific gene regulation and GR's tissue specific function (doi: 10.1101/gr.188581.114, doi: 10.1186/s13059-014-0418-y) and the similarity of AR and GR (half-) sites (DOI 10.1007/s00018-017-2467-3) of which they identify a good number in the GR binding site Motif analysis. As it is an AR-half site motif that overlaps the S1 STAT binding motif (see below) and this is affected or deleted in dN-S3 and dN-S1/3. I do not think it changes the overall conclusions of the manuscript but for completeness it should be addresses in a bit more detail.

Response

We thank the reviewer for the valuable insight into the GR system. We have now analyzed the presence of GR half-site motifs in the *Csn1s2b* regulatory regions and identified one half-site that overlaps with the S1 STAT5 binding site (highlighted in Supplementary Table 3 and lines 116-118). This presence of a half-site motif opens the possibility of direct GR binding to this sequence, possibly cooperatively with STAT5 binding to sites S2 and S3 as well as NFIB. In support of this, inactivation of the STAT5 site S2 results in greatly reduced GR binding despite of the presence of a wt GR half site. The overlap of the GR half site and the 5' part of the S1 GAS motif complicates the situation even further since it should not be possible for two TFs to simultaneously bind to coinciding sites. Thus, it remains to be seen whether GR binds directly to the half-site motif or indirectly through tethering to STAT5 (or both). Even STAT5 might bind indirectly through GR at site S1.

Clear support that GR can bind at STAT5 sites in the absence of GR half-sites (likely through tethering to STAT5) comes from our earlier studies on the *Wap* super-enhancer⁵. In that study, no GR motifs were detected in the vicinity of STAT5 binding sites and deletion of the GAS motifs resulted in the loss of STAT5 binding (as expected) but also in the complete loss of GR and NFIB binding⁵. We added additional information (lines 299-303). While GR binding at mammary enhancers is evident, the overall role of GR is not as the deletion of the GR from the mouse genome does not negatively impact mammary gene expression^{6,7}.

Per suggestion of the reviewer, we analyzed available PR⁸ and ER⁹ ChIP-seq data from mammary tissue from non-parous mice. There was no binding of PR and ER in the *Csn1s2b* upstream enhancer. We added this information in the discussion (lines 302-312).

Some minor points

1) *Line 59 & 87: size/ definition of extend of casein locus ~400kb or ~280 kb?*

Response

The five casein genes are positioned within a ~400kbp locus. We corrected the typo in line 88.

2) *Line 273: “..where it induces gene expression several hundred-fold...” Suggest: ..where it contributes (enables) to a several hundred fold expression induction...*

Response

We corrected this.

3) *Line 275/276: appears to be an incomplete sentence*

Response

We corrected this.

4) *Line 305: “...absence of an overt biding site.” see above: there appears to be an AR (GR) half-site.*

Response

Yes, a half-site (TGTYCY) that can be recognized by several nuclear receptors, including the GR and PR, has been located in the distal enhancer and is part of the STAT5 site S1. We discussed this (lines 117, 129, 302-304).

5) *Line: 308: conceptually, I take issue calling it “strength” of an enhancer, IMO the*

enhancer enables/mediates (possibly in conjunction with the EI in pregnancy) the exceptionally high transcription of the csn1s2b gene to only initiate/occur during lactation while the other genes in this gene cluster initiate such high transcription at an earlier point (pregnancy-early lactation)

Response

We edited this statement and incorporated the reviewer's suggestion (lines 314 – 316).

6) Line321: "...Three STAT binding in..." seems incomplete: binding sites?

Response

We completed the sentence (Line 331).

7) Discussion regarding GR and PR: there is virgin PR Chip-seq data and I believe there is extensive overlap with GR bound sites in lactation. (PR repression might be another part of the puzzle)

Response

Yes, PR ChIP-seq data had been generated from mammary tissue from non-parous mice injected with progesterone⁸. We have now analyzed these data and no binding in the distal enhancer was detected. We also analyzed estrogen receptor (ER) ChIP-seq data from mammary tissue from non-parous mice⁹. We added this information (lines 305-312).

1. Hennighausen, L. *et al.* High-affinity binding site for a specific nuclear protein in the human IgM gene. *Nature* **314**, 289-92 (1985).
2. Gronostajski, R.M., Nagata, K. & Hurwitz, J. Isolation of human DNA sequences that bind to nuclear factor I, a host protein involved in adenovirus DNA replication. *Proc Natl Acad Sci U S A* **81**, 4013-7 (1984).
3. Gronostajski, R.M., Adhya, S., Nagata, K., Guggenheimer, R.A. & Hurwitz, J. Site-specific DNA binding of nuclear factor I: analyses of cellular binding sites. *Mol Cell Biol* **5**, 964-71 (1985).
4. Robinson, G.W. *et al.* Coregulation of genetic programs by the transcription factors NFIB and STAT5. *Mol Endocrinol* **28**, 758-67 (2014).
5. Shin, H.Y. *et al.* Hierarchy within the mammary STAT5-driven Wap super-enhancer. *Nat Genet* **48**, 904-911 (2016).
6. Reichardt, H.M. *et al.* Mammary gland development and lactation are controlled by different glucocorticoid receptor activities. *Eur J Endocrinol* **145**, 519-27 (2001).
7. Kingsley-Kallesen, M. *et al.* The mineralocorticoid receptor may compensate for the loss of the glucocorticoid receptor at specific stages of mammary gland development. *Mol Endocrinol* **16**, 2008-18 (2002).
8. Lain, A.R., Creighton, C.J. & Conneely, O.M. Research resource: progesterone receptor targetome underlying mammary gland branching morphogenesis. *Mol Endocrinol* **27**, 1743-61 (2013).

9. Palaniappan, M. *et al.* The genomic landscape of estrogen receptor α binding sites in mouse mammary gland. *PLoS One* **14**, e0220311 (2019).

REVIEWERS' COMMENTS

Reviewer #2 (Remarks to the Author):

The Authors have addressed all concerns.

2 minor typo's in added text

Line 314: "reveal ant binding"... any binding

Line319 "..enhancers active other casein genes".... activate